# Assessing Multilateral Development Bank ESG Safeguard Integration with International Sustainability Ratings

**Damián Rodríguez Estévez [1,\*] and Rosa María Arce Ruíz [2]**

1   Civil Engineering Systems, ETSICCP, Polytechnic University of Madrid, Campus Prof. Aranguren, No. 3, 28040 Madrid, Spain
2   TRANSyT-UPM Transport Research Centre, 28040 Madrid, Spain; rosa.arce.ruiz@upm.es
\*   Correspondence: damian.rodriguez@alumnos.upm.es; Tel.: +34-673-058-046

**Abstract:** In an era where sustainability is paramount, this study critically assesses how multilateral development banks (MDBs) integrate internationally recognized sustainability indicators into their ESG safeguard policies. MDBs have historically incorporated policies to manage environmental and social risks in project financing; yet, protections against negative impacts in developing countries often remain insufficient. On the other hand, several infrastructure sustainability rating systems have been established around the world in recent decades due to economic growth and the importance of controlling environmental impacts associated with the construction sector. The purpose of this study was to analyze whether and how the indicators that these internationally recognized systems use to rate whether a project is sustainable are integrated into these safeguards by using several methodologies, including an analysis of existing documentation, a high-level matrix, and qualitative methods based on co-occurrences using specialized "atlas ti" software. The results show that MDBs' coverage of financial, governance, and country risks lacks the sustainability focus found in these rating systems. Therefore, this study that concludes MDB safeguards must evolve, balancing comprehensive sustainability parameters and detailed management guidelines and addressing impacts beyond statutory frameworks to encourage stakeholder engagement for more sustainable infrastructure projects.

**Keywords:** ESG safeguards; multilateral development banks; sustainable infrastructure; rating tools

## 1. Introduction

Multilateral development banks (MDBs) and other development finance institutions can play an important role in helping governments create an effective enabling environment for sustainable infrastructure through technical assistance and project preparation as well as in mobilizing private action [1]. To achieve this goal, these organizations have relied on the implementation of safeguarding policies accompanied by a deliberate effort to harmonize them while considering regional and stakeholder differences [2].

The development of these safeguard systems began in the 1990s as a tool for safeguarding sustainability in infrastructure projects. Since then, these systems have been actively updated in an attempt to bring together as many potentially affected social, environmental, and economic areas as possible [3]. Some authors agree that there has been little academic analysis of how these safeguard systems cover all the mandatory credits or parameters of international certification systems, but there is a broad consensus that they have improved in their comprehensiveness over the years [4]. These systems are incorporated into the governance of multilateral development banks as part of the tools available for decision-making, defining a set of environmental and social requirements that potential borrowers must meet during project design and construction. However, although they incorporate these requirements, they lack a system of clear and standardized indicators that can group together each of the areas they seek to ensure. As far as the different international

certification systems are concerned, there is no standardized or commonly agreed-upon methodology that offers a reliable measurement of sustainability throughout the life cycle of infrastructure [5]. Multilateral entities that provide funding for borrowers to undertake projects require guidelines and practical tools that allow them to monitor the impact generated during all stages of a project—tools that are also necessary for the governments and authorities involved [6,7]. This research seeks to contribute knowledge to a hitherto unresearched approach that seeks to identify the extent of the relationship between multilateral development banks' safeguard systems and internationally recognized infrastructure sustainability certification systems. This is an opportunity to carve a new niche into the academic landscape by departing from the existing literature, which mainly focuses on the role of multilateral development bank (MDB) environmental, social, and governance (ESG) safeguard policies in advancing project sustainability and the use of ENVISION and CEEQUAL for tracking sustainable progress in ongoing projects. Our investigation explores uncharted territory. This paper's novel analysis of the integration of credits and monitoring indicators from these international certifications into MDB safeguard policies presents a significant learning opportunity for the academic community and policymakers. By dissecting the extent and manner of this integration, this study lays the groundwork for refining these policies toward more effective and actionable strategies in the construction phases of projects. Ultimately, it ensures that projects not only adhere to best practices but are also directed by a robust framework equipped with clear and measurable indicators, marking a step toward operational excellence and enhanced sustainability outcomes for projects financed by these banks.

The main objective of this study was to answer the following question: Does MDB safeguard policies comply with all the items and indicators monitored in the credits of the international sustainability certificates? It also aimed to compare the safeguard policy systems available for each selected MDB in order to assess which MDB has the highest level of sustainability area integration, indicated by recognized infrastructures of sustainability certification schemes. This research is not an attempt to validate whether the current safeguards are individually successful but rather to ascertain the degree to which these policies adopt the sustainability approach of these international certificates. The specific objectives of this research were to, first, benchmark the performance of these safeguards in measuring sustainability against the criteria of the proposed certification schemes; secondly, provide a comparative analysis of the elements and scopes identified as priorities by the certification schemes and the extent to which they are featured in safeguards; and, third, identify challenges to improving the sustainability shielding of infrastructure projects financed by these bodies in the future. To this end, this work provides a broad overview of the state of the art of sustainability measurement. Finally, two major research needs are proposed to improve the integration of all the necessary aspects to be included in safeguards and covered during the design and construction of infrastructure projects.

## 2. Literature Review

### 2.1. ESG Safeguard Systems

To ensure sustainable development, development finance institutions have developed a set of original environmental and social safeguard policy documents that "describe how the Bank will address the environmental and social impacts of its projects" [8]. (See Figure 1). These safeguards are defined by each entity's legal framework and mandate and are tailored to the nature of the projects they implement and the local sensitivities of each region. MDBs like the World Bank Group have been pioneers in implementing robust environmental and social safeguard systems [9]. Therefore, the different existing safeguard policies exhibit great differences in approach and scope [10].

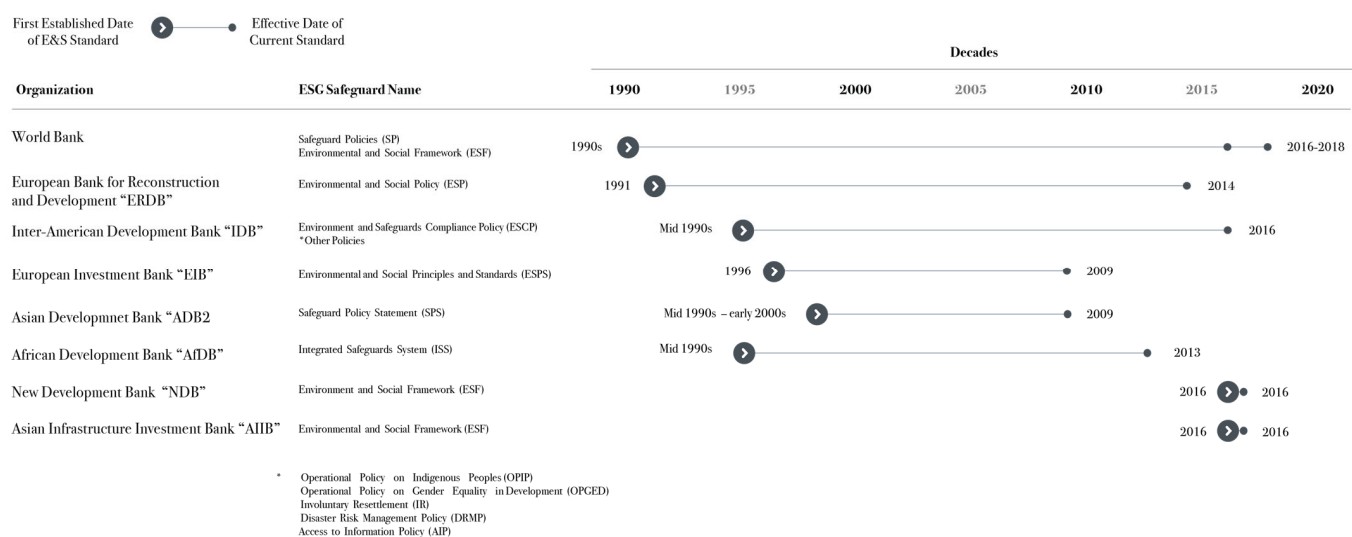

**Figure 1.** ESG safeguard timeline. Own elaboration based on various documents [8,10].

Research has shown that ESG factors can significantly impact bank performance and risk-taking behavior [11]. Banks that prioritize ESG considerations tend to exhibit better risk management practices and create long-term value for stakeholders. Additionally, studies have highlighted the importance of sustainability reporting in the banking sector, emphasizing transparency regarding ESG issues [12]. By disclosing ESG information, banks can enhance their reputation and demonstrate their commitment to sustainable practices.

Currently, there is a significant divergence in safeguard policy between development banks and the different regions in which they operate, with Western-backed development banks requiring borrowers to conform to developed country standards, while others, such as banks in China and Brazil, abide by host country standards [13].

Many organizations, due to this lack of real adaptation to the markets in which they operate, are revising and adapting their processes to the new paradigm of the global legal order, and a good example of this is the World Bank's new "Environmental and Social Framework" (ESF) that replaced the old "Safeguard Policies" that had gradually emerged since the 1980s in response to the harmful impacts of World Bank-financed investment projects [14]. The European Union has indeed made significant progress in advancing legislation related to environmental, social, and governance (ESG) issues. One notable directive is the "Non-Financial Reporting Directive" introduced in 2014, which mandates that large companies with over 500 employees disclose ESG information [15]. This directive has influenced the value relevance of ESG disclosure as firms voluntarily providing non-financial reports before its application have seen positive outcomes [16]. Additionally, the European Commission has deployed various financial instruments and funding to address energy poverty, showcasing a commitment to tackling social and environmental challenges [17]. Furthermore, the EU has endorsed integrated coastal zone management (ICZM) as a tool for managing complex coastal issues, highlighting a proactive approach to coastal governance and risk management [18]. These legislative efforts demonstrate the EU's dedication to promoting sustainability, transparency, and accountability in various sectors, contributing to a more sustainable and responsible business environment.

The analysis and design of the most sustainable scenarios for an infrastructure project are usually undertaken early in the project cycle, not least because it is at this stage that there is an opportunity to weigh up different options and consider alternative materials or even a different design, including the risks to which a project may be exposed throughout its life cycle. Consequently, multilateral development banks, which aim to support investment in sustainable infrastructure, could help through their safeguards to ensure the sustainability of infrastructure [1].

It is important to mention that the alignment of MDBs to this taxonomy seems to be a crucial and essential step [19], considering the incorporation of the EU taxonomy as a classification system for economic activities that can be considered environmentally sustainable, as published by the European Commission in March 2020. As authors such as those of [20] warn in their article "Environmental Impact Assessment for transport projects: A review of technical and process-related issues" regarding EIAs, this taxonomy system published by the EU is merely oriented toward the analysis of sustainability from an environmental point of view, so it seems reasonable to argue that both natural and socioeconomic aspects related to infrastructure projects should be integrated more efficiently. Although the proposed taxonomy is only environmentally focused, we believe that it will not take long to integrate a complete taxonomy, not only for environmental but also for social and economic monitoring purposes as well as for all types of infrastructure projects. When the relevant financial market actors, including multilateral development banks, start complying with the standards, the effectiveness of this classification system can be fully assessed [19].

The safeguards designed by MDBs generally do not follow international environmental and social standards but only monitor the minimum requirements among member countries [21]. The two authors of [21] also discuss accountability mechanisms with reference to the sustainability of projects funded by these entities and how they could be more effective. MDBs often play a key role in helping the recipient country improve socioenvironmental planning by establishing or reinforcing environmental and social requirements [22].

The environmental and social safeguard approach used by the major multilateral development banks (MDBs) requires a thorough rethink to address the gaps between project conceptualization and practice [3]. Despite the proven benefits of implementing these widely discussed safeguard policies, some studies emphasize that safeguards serve mainly to protect the banks themselves from criticism; that they do not affect most non-MDB-funded projects; that they are expensive, complicated, and highly bureaucratic; and that they provide no incentive for borrowing countries to improve their systems [23]. According to some authors [24], the integration of these safeguards into projects is not optimal, despite the constant and diverse adaptation of their policies, as these multilateral organizations are constantly undergoing reform, which indicates that there are only "good practices" instead of so-called "best practices". Therefore, there have been many attempts by these organizations to improve or verify their standards by benchmarking against external standards [1].

One of the main issues with measuring the impact of these safeguards is the paucity of open data disclosed by the organizations themselves on the outcome of using these tools. Qualitative scrutiny of disclosure standards, with an emphasis on aspects such as completeness of disclosure, accessibility of information, timeliness of information, and availability of resources, reveals that most development multilaterals operate under the strictest of corporate confidentiality principles, conflicting with the public demand for information [25].

Moreover, recent research in the field of sustainable finance, development outcomes, and ESG impact assessment methodologies has seen significant advancements. Studies such as [26] have focused on sustainable finance initiatives in Japan, highlighting public policy efforts to integrate ESG criteria into financial decision-making and promote sustainable finance [26].

The authors of [27] conducted a bibliometric and content analysis of ESG literature, revealing a growing interest in ESG within the sustainable finance domain, as evidenced by increasing publications and citations [27]. The authors of [28] developed a methodology for assessing transition finance, aligning with the finance for sustainable development approach and emphasizing the role of external finance sources in countries' transitions toward sustainability [28]. These studies collectively contribute to the evolving landscape of sustainable finance, development outcomes, and ESG impact assessment methodologies, providing valuable insights into the intersection of finance, sustainability, and governance practices.

As we have seen, many studies have examined the performance achieved by projects adopting ESG systems, not only in finance or infrastructure projects but also in other more exotic projects such as the work "Terrorist attacks and environmental social and governance performance: Evidence from cross-country panel data" [29], which goes beyond what is understood by the performance of sustainability, investigating the relationship between levels of terrorism and social and environmental performance measured in ESG terms.

### 2.2. Infrastructure Sustainability Certification Schemes

Sustainability assessment is a complex assessment method. It is undertaken to support decision-making and policy in a broad environmental, economic, and social context, and transcends a purely technical/scientific assessment" [30]. Sustainability rating tools (See Table 1) are a response to the need to translate high-level concepts and objectives into practical frameworks for infrastructure owners and design and construction professionals [22].

**Table 1.** Sustainability rating tools.

| Tool | Certifying Body | Sector | Country |
| --- | --- | --- | --- |
| ASCE | American Society of Civil Engineers | All | US |
| CEEQUAL | Institution of Civil Engineers [31] | All | UK |
| ENVISION | Institute of Sustainable Infrastructure (ISI) | All | US |
| IS | Australian Green Infrastructure Council (AGIC) | All | Australia |
| GreenLITES | New York State Department of Transport | Transport | US |
| GREENROADS | University of Washington | Transport | US |
| I-LAST | Illinois Department of Transportation | Transport | US |
| INVEST | Federal Highway Administration (FHWA) | Transport | US |
| STARS | Portland Bureau of Transport | Transport | US |

The construction sector has implemented different sustainability assessment and certification systems due to the need to understand construction from a sustainable development approach [32]. As a result of this effort, there are now a multitude of assessment and certification systems that measure and quantify the sustainability of infrastructure projects. They are often developed by governmental and non-governmental institutions and sometimes in collaboration with academia [33].

Some of the existing sustainability assessment systems are state-level and others are national [34]. These systems use different techniques to determine sustainability, emphasizing different sustainability factors, as proposed in "Toward more sustainable infrastructure" [35]. Based on the scientific community's accepted definition, the scope of sustainability assessments consists of the identification, prediction, and evaluation of the potential impact of different solutions or alternatives through the triple bottom line [36].

The different certification schemes currently recognized are designed to provide guidance, scoring, and potential rewards for using sustainable best practices [37]. Most of these schemes go beyond the existing minimum regulatory requirements imposed by the regulations of the countries where the projects are located.

Although these systems have undergone improvements following various reviews of their scope, they are often criticized because they tend not to fully consider economic and social aspects, to the detriment of environmental issues [38]. Some analysts suggest that infrastructure developers and owners considering the use of such sustainability rating tools should be aware of the focus and bias of the tools available on the market since, if only the requirements proposed by these rating tools are analyzed, the areas of sustainability more relevant to a project and its wider context may be overlooked [39].

Authors such as those of [33] analyze and describe some of the methods specifically designed to assess sustainability, such as rating systems, and conclude that they are incomplete as they do not consider aspects such as cost–benefit analysis, multi-criteria decision analysis, or environmental impact assessment methods. Many of the multilateral agencies and offices that incorporate infrastructure development under their mandates have not yet widely incorporated these rating systems in the evaluation of their projects [40], although they have been using this type of system since 2015 to evaluate their sustainability from a corporate point of view, such as the use of the GRI system, which is practically in general use by the organizations of the United Nations system. As a point to bear in mind, the rating systems are also deficient because they focus on developed economies and do not consider the integration of the local sensitivities and specific characteristics of the countries in which these infrastructures are developed [41].

There are three general infrastructure sustainability assessment and certification systems applicable to any type of infrastructure that assess projects following the principles of sustainability (ENVISION ISI, 2012; Civil Engineering Environment Quality "CEEQUAL", BRE Group, 2015; and Infrastructure Sustainability "IS" ISCA, 2012), although there are others that integrate infrastructures more oriented toward transport or building and that are not included in this study due to the specificity of their scope. There are elements in common between most of the systems analyzed for the assessment of the sustainability of infrastructure projects and that are presented under the pillars of Management, People, Resources, and the Natural World, although in some there are also more specific ones [42]. Although all these elements consider the most important aspects of sustainability, focusing on specific criteria, there are no major differences between them with respect to the requirements for an infrastructure project to be considered sustainable [41].

If we consider the nature of the assessment system itself from the point of view of the methodology used, it can be determined that most systems are usually associated with a common metric, usually called points or credits [5]. These rating tools are generally structured around a list of elements/criteria organized into main themes/categories, such as location, linkage, planning, sustainability, energy and water resources, materials, infrastructure, waste management, transport, land-use planning, social and economic well-being, and innovative design and technology [43], but, even so, there is no uniformity or standardization of criteria.

Some ranking systems such as CEEQUAL weigh each best practice equally, while others such as ENVISION set different levels of importance for each best practice. In conclusion, it should be mentioned that these systems, in terms of development history, strategy choices, evaluation structure, evaluation criteria, and local benchmarks, are very diverse [43].

Taking into account the analysis in "Sustainable Assessment of Transport Infrastructures Projects: A Review of Existing Tools and Methods" [5], four basic reasons for the desirability of these models are listed:

i.     They provide a common metric for the whole range of sustainable solutions;
ii.    They measure sustainability and, therefore, make it manageable;
iii.   They allow direct communication of sustainability goals, efforts, and achievements;
iv.   They provide a reasonable context in which designers, contractors, and material suppliers can be innovative in their solutions [44].

Some comparative studies reflecting the effectiveness and applicability of these systems can be found in the reference literature [45].

So, it has been highlighted that there are no previous publications that analyze what has been discussed in this paper. Of course, there are publications that analyze how MDBs, through their ESG safeguard policies, contribute to achieving the sustainability of projects and how ENVISION and CEEQUAL contribute to monitoring the progress of the sustainability of already implemented projects, as has been highlighted in this literature review, but there are none that analyze what is proposed in this paper, and that is why there is such a gap. By analyzing how and how many of these issues through the indicators

used by these international certificates are already integrated into safeguards policies, it will be much easier to move toward revising these policies and orienting them toward more operational and impactful approaches to projects during construction, ensuring that projects are not only subject to good practice but also to a structured framework with clear indicators.

## 3. Methodology

The aim of this study was to determine the degree to which the ESG safeguards used by MDBs in the design and implementation phase of infrastructure projects of all types are related to the credits that two internationally recognized and prestigious international systems use to determine the degree of sustainability of infrastructure projects. In the first stage, by way of bibliographic analysis, all the published documentation on sustainability certification systems was reviewed, as well as the MDBs' safeguard policies, in order to understand their scope. In the second or selective analysis stage, a series of variables were utilized to select the entities to be reviewed. In the third or qualitative analysis stage, two different tools were used. Firstly, a high-level matrix was created, which was a simplified version of the "ecosystem services matrix" [46], in which the relationship between safeguards and sustainability certification systems was analyzed and contrasted in a second phase of this stage by means of qualitative analysis using atlas ti software. In this last stage of qualitative analysis, it was necessary to define the indicators that allowed the coding and subcoding of the documents to be studied. These indicators, as well as the codes and subcodes generated, are available upon request.

The following Figure 2 shows the different stages and research methods undertaken in this study.

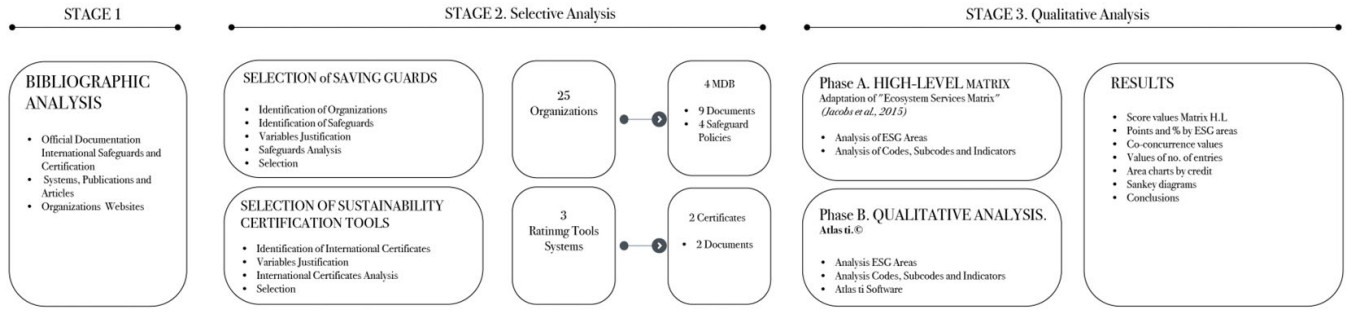

**Figure 2.** Stages of the methodology applied for this research. Own elaboration [46].

Figure 3 sets out the thematic areas and parameters that ESG safeguards seek to control in their projects and correlates directly with the codes in Table 4, categorized by the three dimensions of sustainability [29]. Each point in Table 4, such as, for example, "Pollution Prevention and Abatement" (MA.1) or "Indigenous Peoples" (S.1), is reflected in a specific code, providing a structured framework for systematic reference and management. In relation to this, Figure 4 details the essential components in the sustainability certification schemes, which are coded in Table 7, e.g., for the CEEQUAL scheme, "Flooding and Surface Water Run-Off" (2.2), or Welfare (QL1) for ENVISION. The purpose of these 2 figures and the tables was to compose a system of codes classified by area and parameter that allowed us to assess their interrelation both in the high-level matrix and in the qualitative analysis through co-occurrences.

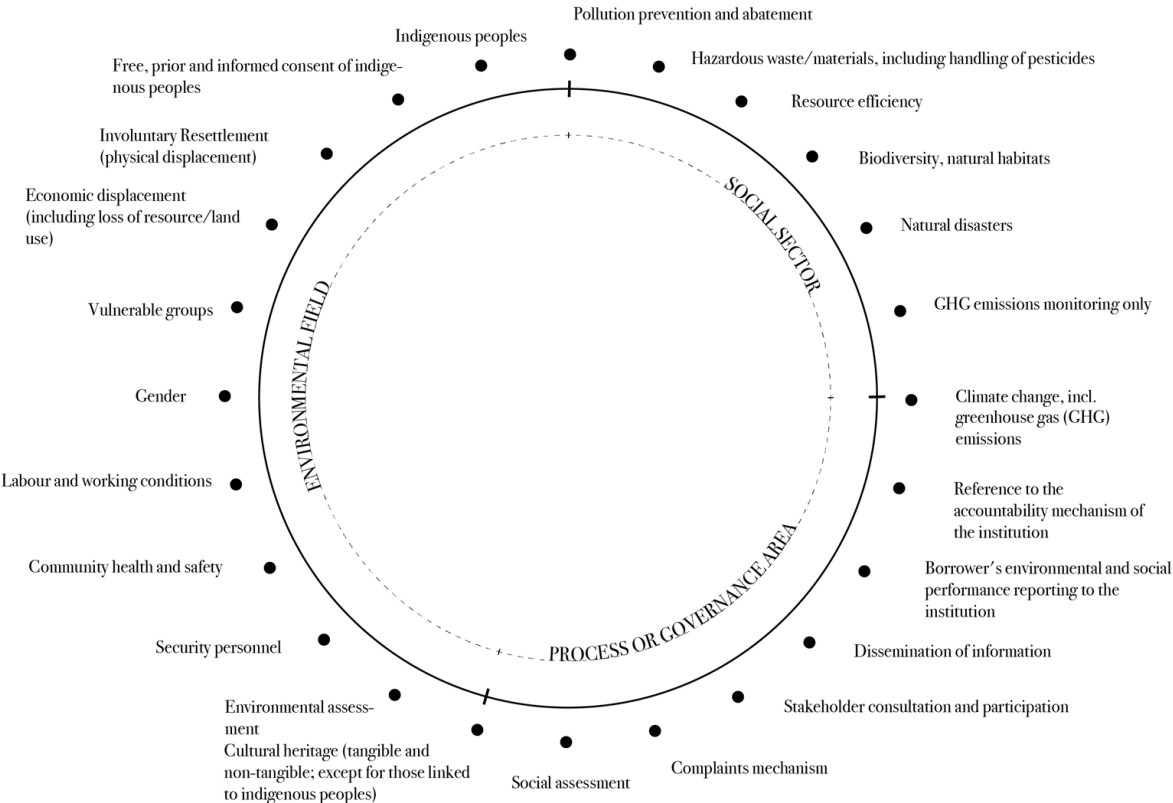

**Figure 3.** Areas and parameters controlled by ESG safeguards. Own elaboration based on several documents [10].

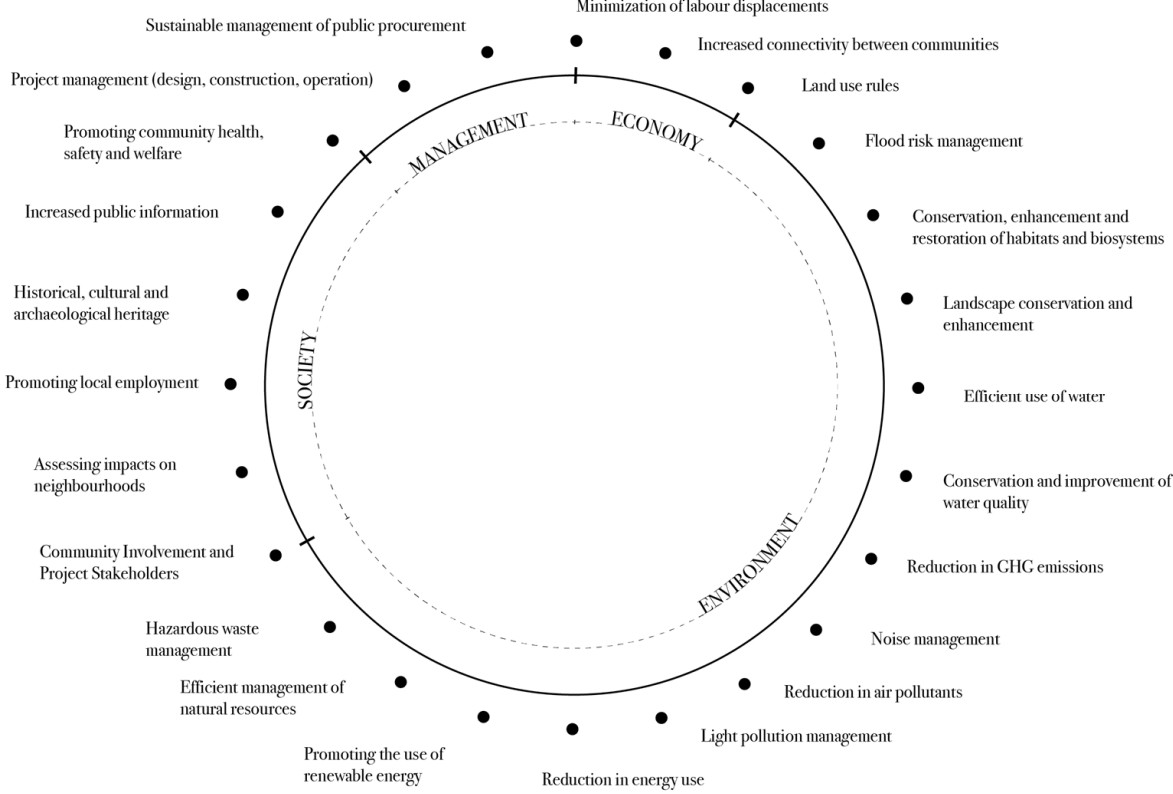

**Figure 4.** Parameters and aspects of international sustainability certification systems. Own elaboration based on main criteria used in the evaluation of projects by the rating systems of [37].

## 4. Analysis

### 4.1. Identification of Multilateral Organizations: Criteria to Be Benchmarked

Even though, according to the latest classification of MDBs, there are 25 organizations of a development banking nature [10], the focus and research was on analyzing the organizations that could cover each one of the parameters defined in the following list of variables.

(a)   The MDB has a global or regional scope.

(b)   The guidelines of the organization's mandates include the promotion of sustainable economic development and the support of regional cooperation.

(c)   The organization's credit rating is AAA.

(d)   The financial instruments with which the organization's finance infrastructure projects include loans, lines of credit, and technical assistance during the design and implementation phase of projects.

(e)   The organization invests more than 40% of its total disbursements in infrastructure development and specifically in transport, energy, and communications, according to the sectoral breakdown of MDB disbursements concessional and non-concessional, 2015. (See Table 2).

**Table 2.** Selected variables covered by analyzed MDBs. Own elaboration.

| | Variables | | | | |
|---|---|---|---|---|---|
| **MDB** | **a** | **b** | **c** | **d** | **e** |
| Asian Development Bank (ADB) | Regional | ● | ● | ● | ● |
| African Development Bank [47] | Regional | ● | ● | ● | ● |
| Asian Infrastructure Investment Bank (AIIB) | Regional | ● | ● | ● | – |
| Inter-American Development Bank (IDB) | Regional | ● | ● | ● | – |
| European Bank for Reconstruction and Development (EBRD) | Regional | ● | ● | ● | ● |
| International Finance Corporation (IFC) | Regional | ● | – | ● | – |
| World Bank (WB) | Global | ● | – | ● | – |

Data extracted from "Guide to Multilateral Development Banks 2018". ● = Complies; – = Not compliant, partially compliant, or no published data.

### 4.2. Selection of Bodies to Be Analyzed

We analyzed seven international organizations and, after the process of analysis based on the variables set out above, we determined that the number of entities that fully complied with the variables was four.

The following Table 3 shows the analyzed entities as well as the documentation relating to the safeguards included in the object of study.

**Table 3.** MDBs selected for assessment.

| Org | Date ESG First Established | Current ESG Safeguard Policies | Date of Entry | No. of Documents | No. of Words |
|---|---|---|---|---|---|
| ADB | Mid-1990s–early 2000s | Safeguard Policy Statement SPS | 2009 | 1 | 40,949 |
| AFDB | Mid-1990s | Integrated Safeguards System ISS | 2013 | 1 | 26,432 |
| IDB | Mid-1990s | Environment and Safeguards Compliance Policy ESCP | 2016 | 6 | 25,294 |
| EBRD | 1991 | Environmental and Social Policy ESP | 2014 | 1 | 33,573 |

### 4.3. Identification of Codes, Subcodes, and Reference Indicators in ESG Safeguards

Once we had selected the entities as well as the documents that made up their safeguards, as shown in Table 3 above, we defined the indicators that characterized each of the codes and subcodes of the safeguards. In this process, we were guided by the areas proposed by the guide [10], which reflects that these areas were those that were most com-

monly found in a standardized manner in practically all safeguard policies, regardless of the document in which they were found and whether they were explicitly or implicitly defined.

As can be seen in the following Table 4, these thematic areas were grouped according to the three axes of [10]—the backbone of ESG safeguards—which is environment, society, and governance. This table shows the six subcodes into which the environmental thematic area was divided; the ten that applied to the social area, this being the one that contributed most to this classification; and, finally, the six that made up the governance area, as indicated in the study [10].

**Table 4.** ESG safeguard codes and subcodes. Own elaboration.

| Codes | No. | Subcodes—OVE 2018 Thematic Areas |
|---|---|---|
| ENVIRONMENT | MA.1 | PREVENTION OF AND REDUCTION IN POLLUTION |
| | MA.2 | WASTE/HAZARDOUS MATERIALS, INCLUDING PESTICIDE MANAGEMENT |
| | MA.3 | RESOURCE EFFICIENCY |
| | MA.4 | BIODIVERSITY, NATURAL HABITATS |
| | MA.4 | NATURAL DISASTERS |
| | MA.5 | CLIMATE CHANGE, INCL. GREENHOUSE GAS (GHG) EMISSIONS |
| | MA.6 | TRANSBOUNDARY ISSUES |
| SOCIAL | S.1 | INDIGENOUS PEOPLES |
| | S.2 | FREE, PRIOR AND INFORMED CONSENT OF INDIGENOUS PEOPLES |
| | S.3 | INVOLUNTARY RESETTLEMENT (PHYSICAL DISPLACEMENT) |
| | S.4 | ECONOMIC DISPLACEMENT (INCLUDING LOSS OF RESOURCES/LAND USE) |
| | S.5 | VULNERABLE GROUPS |
| | S.6 | GENDER |
| | S.7 | LABOUR AND WORKING CONDITIONS |
| | S.8 | COMMUNITY HEALTH AND SAFETY |
| | S.9 | SECURITY PERSONNEL |
| | S.10 | CULTURAL HERITAGE (TANGIBLE AND INTANGIBLE) |
| GOVERNANCE | G.1 | ENVIRONMENTAL ASSESSMENT |
| | G.2 | SOCIAL ASSESSMENT |
| | G.3 | STAKEHOLDER CONSULTATION AND PARTICIPATION |
| | G.4 | GRIEVANCE MECHANISM |
| | G.5 | ENVIRONMENTAL AND SOCIAL PERFORMANCE REPORTING BY BORROWER |
| | G.6 | REFERENCE TO THE INSTITUTION'S ACCOUNTABILITY MECHANISM |

*4.4. Identifying Sustainability Tools as a Reference for Analysis*

The focus of the research was on the mapping and detailed analysis of existing sustainability measurement and reporting methods in the market. Although there are hundreds of sustainability measurement systems in use around the world, ranging from simple spreadsheet-based approaches to cloud-based systems, the selection of the two systems to be analyzed was based on the fulfillment of four criteria:

- A generalist scope for all types of infrastructures;
- International recognition;
- Widely used, global systems;
- Accessibility to documentation and datasets to allow for detailed analyses.
- From the above variables, the certificates included in Table 5 below were included.

Taking as a reference the variables previously determined as a starting point for selecting the rating systems and the information in Table 5, two of the three proposed systems were identified due to the ease of obtaining the reference documentation and its availability in the various sources consulted. For this reason, the analysis focused on the following systems, as shown in Table 6 below.

**Table 5.** Sustainable infrastructure rating tools: main features. Own elaboration.

| Rating Tool | ENVISION | CEEQUAL | Infrastructure Sustainability |
|---|---|---|---|
| Launch of the first version of the tool | 2012 | 2003 | 2012 |
| Country of origin | USA | United Kingdom | Australia |
| Life cycle phases covered by the tool | Planning and design | Planning and design | Design, construction, and operations |
| Sustainability issues addressed | 1. Quality of life<br>2. Leadership<br>3. Resource allocation<br>4. The natural world<br>5. Climate change and risk | 1. Pollution<br>2. Land use and ecology<br>3. Resources<br>4. Transport<br>5. Communities and stakeholders<br>6. Management<br>7. Resilience<br>8. Landscape and historic environment | 1. Management and governance<br>2. Resource use, materials, and waste<br>3. Ecology<br>4. People and places<br>5. Innovation |
| Categories | 5 | 8 | 5 |
| Subcategories | 14 | 30 | 15 |
| Certificate levels | -Platinum<br>-Gold<br>-Silver<br>-Bronze | -Excellent<br>-Very good<br>-Good<br>-Pass | -Leader<br>-Excellent<br>-Recommended |
| Version used for analysis | 3 | 6 | 2.1 |
| Open data accessibility | YES | NO | NO |

**Table 6.** Sustainable infrastructure rating tools selected.

| System | Version | Year of Publication | No. of Words |
|---|---|---|---|
| ENVISIÓN | V.3 | 2018 | 91,328 |
| CEEQUAL | V.6 | 2020 | 67,680 |

*4.5. Identification of Codes, Subcodes, and Reference Indicators of the Selected Systems*

The indicators identified for each primary code (theme) and subcode (subtheme) and the number of indicators of the two systems to be compared are illustrated in Table 7 below. A more detailed explanation of the indicators used is available upon request.

**Table 7.** CEEQUAL and ENVISION codes, subcodes, and indicators. Own elaboration.

| System | Code | No. | Subcode | No. of Indicators |
|---|---|---|---|---|
| **CEEQUAL** | MANAGEMENT | 1.1 | SUSTAINABILITY LEADERSHIP | 8 |
| | | 1.2 | ENVIRONMENTAL MANAGEMENT | 8 |
| | | 1.3 | RESPONSIBLE CONSTRUCTION MANAGEMENT | 3 |
| | | 1.4 | SOCIAL GOVERNANCE OF STAFF AND THE SUPPLY CHAIN | 4 |
| | | 1.5 | WHOLE-LIFE COSTING | 1 |
| | RESILIENCE | 2.1 | RISK ASSESSMENT AND MITIGATION | 4 |
| | | 2.2 | FLOODING AND SURFACE WATER RUN-OFF | 3 |
| | | 2.3 | FUTURE NEEDS | 2 |

**Table 7.** *Cont.*

| System | Code | No. | Subcode | No. of Indicators |
|---|---|---|---|---|
| CEEQUAL | COMMUNITIES AND STAKEHOLDERS | 3.1 | CONSULTATION AND ENGAGEMENT | 13 |
| | | 3.2 | WIDER SOCIAL BENEFITS | 6 |
| | | 3.3 | WIDER ECONOMIC BENEFITS | 3 |
| | LAND USE AND ECOLOGY | 4.1 | LAND USE AND LAND VALUE | 8 |
| | | 4.2 | LAND CONTAMINATION AND REMEDIATION | 4 |
| | | 4.3 | BIODIVERSITY PROTECTION | 6 |
| | | 4.4 | BIODIVERSITY CHANGE AND ENHANCEMENT | 4 |
| | | 4.5 | LONG-TERM MANAGEMENT OF BIODIVERSITY | 2 |
| | LANDSCAPE AND HISTORIC ENVIRONMENT | 5.1 | LANDSCAPE AND VISUAL IMPACTS | 7 |
| | | 5.2 | HERITAGE ASSETS | 5 |
| | POLLUTION | 6.1 | WATER POLLUTION | 5 |
| | | 6.2 | AIR, NOISE, AND LIGHT POLLUTION | 8 |
| | RESOURCES | 7.1 | RESOURCE EFFICIENCY STRATEGY | 3 |
| | | 7.2 | REDUCTION IN CARBON EMISSIONS OVER A LIFETIME | 4 |
| | | 7.3 | ENVIRONMENTAL IMPACT OF CONSTRUCTION PRODUCTS | 3 |
| | | 7.4 | CIRCULAR USE OF CONSTRUCTION PRODUCTS | 6 |
| | | 7.5 | RESPONSIBLE SOURCING OF CONSTRUCTION PRODUCTS | 6 |
| | | 7.6 | CONSTRUCTION WASTE MANAGEMENT | 5 |
| | | 7.7 | ENERGY USE | 4 |
| | | 7.8 | WATER USE | 7 |
| | TRANSPORT | 8.1 | TRANSPORT NETWORKS | 5 |
| | | 8.2 | CONSTRUCTION LOGISTICS | 5 |
| ENVISION | QUALITY OF LIFE | QL1. | WELFARE | 6 |
| | | QL.2 | MOBILITY | 3 |
| | | QL.3 | COMMUNITY | 4 |
| | LEADERSHIP | LD.1 | COLLABORATION | 4 |
| | | LD.2 | PLANNING | 5 |
| | | LD.3 | ECONOMICS | 6 |
| | RESOURCE ALLOCATION | RA.1 | MATERIALS | 5 |
| | | RA.2 | ENERGY | 4 |
| | | RA.3 | WATER | 4 |
| | NATURAL WORLD | NW1 | SITE | 5 |
| | | NW2 | CONSERVATION | 6 |
| | | NW3 | ECOLOGY | 6 |
| | CLIMATE AND RESILIENCE | CR.1 | EMISSIONS | 3 |
| | | CR2. | RESILIENCE | 6 |

### 4.6. Selection of Methods for Analysis

As presented in Figure 2, two tests were designed and developed to address the research hypothesis. To do this, a number of qualitative methods were used, which are outlined below.

The first technique applied was the use of a high-level matrix mapping methodology that compared the ESG categories of the safeguards of each of the MDBs with the subject matter analyzed using the two systems. The full data and analytical matrices and data records are available from the authors upon request as space is limited in this article and, therefore, only high-level summaries are included.

The second method used was a qualitative textual data mining/analysis technique to identify patterns of intertextual co-occurrence of significance [48]. Both methods used the framework of thematic areas covered by multilateral development bank safeguard policy frameworks [10] to structure and prioritize topics of value for analysis.

### 4.6.1. High-Level Analytical Matrix Linking the Safeguard Areas

The method for constructing high-level associations between the areas covered by the safeguards and the ENVISION and CEEQUAL materiality themes was a simplified and adapted version of the "ecosystem services matrix" [49]. This approach involves the construction of a tabular format to test the strength of linkages across three dimensions and then uses expert panels to test the strength of connection points. This part of the test was limited to the authors' input, so more experts would have been required to further stabilize the results.

The high-level matrix evaluation team assigned scores to a total of 219 indicators, broken down into 67 for ENVISION (spread over the 5 codes and 14 sub-codes) and 152 for CEEQUAL (spread over the 8 codes and 29 sub-codes), defined in Table 7, applying the following scoring scheme:

- Fully Covered (1 point): 1 point was assigned to the indicators of the certification systems that were fully aligned with the objectives and approaches of the subject areas that integrated the different safeguards policies defined in Figure 3. An indicator with this score indicated full and aligned integration with the safeguard.
- Partially Covered (0.5 points): 0.5 points were assigned to indicators that met some, but not all, of the objectives of the thematic area. An indicator in this category indicated that, although partly recognized and addressed, there is still room for more comprehensive coverage.
- Not Covered (0 points): This was assigned to indicators without any focus or content in the safeguards thematic areas. An indicator with a score of zero highlighted opportunities to incorporate new elements in future revisions of certification systems.

However, the technique was used to build an initial composite measure, such as identifying key indicator areas and primary "co-occurrence points" in the samples that could be used in the second phase with the qualitative analysis methodology.

### 4.6.2. Detailed Text Mining Analysis to Establish the Links between CEEQUAL, ENVISION, and ESG Safeguards

The method chosen for the detailed analysis was qualitative textual text mining analysis. With advances in software solutions, text mining is used as a methodology for social scientists in supporting text analysis as it offers the ability to manage and quantify huge amounts of data over a very short time. It is used in academic disciplines such as economics [50], political science [51], and sociology [52]. The specific technique used for this study was code, subcode, and indicator recognition, which provided a statistical technique for capturing "indicator" keywords within the content analysis [53]. This required a coding framework that was constructed from the three thematic areas of the ESG safeguards, the five ENVISION credits, and the eight CEEQUAL credits.

The coding of the documents was carried out based on the indicators that best identified the thematic area of each of the measurement systems and were identified and selected by the authors of this article.

The advanced technique of keyword comparison between texts was first defined by the philosopher and historian Foucault (1973) [48], who identified intertextual patterns that could determine the answers to social science questions. To identify intertextual patterns, text mining requires a series of social science questions and a hierarchy model or "tree map", which, in this case, used the framework of ESG safeguard areas to link key information codes to the specific subcodes and indicators that were associated with these areas.

The qualitative analysis of the words was made possible by a specialized software tool, "atlas ti", qualitative data analysis software that allows for the rapid analysis of large amounts of data.

An objective was that the qualitative research or analysis conducted with the second technique would validate the high-level analytical matrix of the first analysis, providing evidence as to how the ESG safeguards integrate the different subject matter or control credits of the systems used to measure sustainability. The aim of test two was to determine

the degree of semantic or qualitative co-occurrence and which credits were most integrated into these ESG safeguard systems.

## 5. Results

### 5.1. High-Level Matrix Results

By applying the high-level matrix defined in the previous point, we obtained a data matrix large enough to be consolidated as an attached document in point 10 of the appendix of this article. The first results obtained are shown in the following summary Table 8, which illustrates the detailed scores from the high-level matrix by areas and credits, providing an in-depth view of how each bank or financial institution handles the various credits within their ESG safeguard policies.

**Table 8.** High-level matrix scores by areas and credits.

| | ADB | | | IDB | | | EBRD | | | AFDB | | |
|---|---|---|---|---|---|---|---|---|---|---|---|---|
| **ENVISION** | **E** | **S** | **G** | **E** | **S** | **G** | **E** | **S** | **G** | **E** | **S** | **G** |
| QUALITY | 12 | 24.5 | 5 | 12 | 12.5 | 3.5 | 9 | 31 | 5 | 6 | 21 | 4 |
| LEADERSHIP | 1.5 | 16.5 | 15 | 1.5 | 9 | 11 | 1.5 | 20 | 16 | 1.5 | 17 | 13 |
| RESOURCE ALLOCATION | 12.5 | 1.5 | 1 | 12.5 | 0 | 1 | 12.5 | 3 | 1 | 12.5 | 2 | 1 |
| NATURAL WORLD | 19 | 0 | 1 | 19 | 0 | 1 | 19 | 0 | 1 | 19 | 0 | 1 |
| CLIMATE AND RISK | 8 | 1 | 8 | 12 | 1 | 6.5 | 12 | 0.5 | 8 | 12 | 0 | 8 |
| | **ADB** | | | **IDB** | | | **EBRD** | | | **AFDB** | | |
| **CEEQUAL** | **E** | **S** | **G** | **E** | **S** | **G** | **E** | **S** | **G** | **E** | **S** | **G** |
| MANAGEMENT | 47.5 | 13 | 41 | 46.5 | 2.5 | 30 | 49 | 20 | 41 | 46 | 16.5 | 33 |
| RESILIENCE | 26 | 0 | 0 | 29.5 | 0 | 0 | 32.5 | 0 | 0 | 32.5 | 0 | 0 |
| COMMUNITIES AND STAKEHOLDERS | 0 | 97 | 38.5 | 0 | 43.5 | 36 | 0 | 81.5 | 38.5 | 0 | 73.5 | 36.5 |
| LAND USE AND ECOLOGY | 57 | 0 | 3 | 54.5 | 0 | 0 | 59.5 | 0 | 3 | 57 | 0 | 0 |
| LANDSCAPE AND HISTORIC ENVIRONMENT | 5.5 | 23.5 | 39.5 | 5.5 | 21 | 30 | 5.5 | 37 | 39.5 | 5 | 37 | 35.5 |
| POLLUTION | 31.5 | 5 | 34 | 31.5 | 5 | 18 | 31.5 | 5 | 34 | 26.5 | 5 | 26 |
| RESOURCES | 60 | 5.5 | 20 | 57 | 4 | 17 | 60 | 3 | 20 | 60 | 3 | 20 |
| TRANSPORT | 7 | 19 | 34 | 7 | 7 | 10 | 7 | 9 | 34 | 0 | 9 | 34 |

The visual representation provided by the heatmap serves as an analytical tool, offering an immediate, intuitive grasp of the comparative performance of the institutions under consideration—namely ADB, IDB, EBRD, and AFDB—across a spectrum of environmental and governance parameters. This visual format utilizes a color gradient to symbolize the range of scores, with darker hues corresponding to higher scores and lighter ones to lower scores, creating a visually stratified assessment.

On the basis of the average values shown in Table 9, the safeguards were better adapted to the ENVISION certification system than to that of CEEQUAL with regard to the environmental themes, where a difference of 2.2% was reached. This was in contrast to the themes in the area of governance or processes, where the safeguards were better adapted to CEEQUAL, reaching a difference of 10.48%, which was very significant. Finally, it was again in the social area where the safeguards demonstrated a greater adaptation to the ENVISION credits, with a difference of 8.22% from CEEQUAL. As discussed in the previous paragraph in relation to the results achieved, Table 9 below presents a percentage summary of the results obtained through the high-level matrix analysis, illustrating how the sustainability certification credits were distributed among the ESG thematic areas (environmental, social, and governance) in each assessed institution.

**Table 9.** Distribution of credits of the certification systems by ESG safeguard area. Own elaboration.

|  |  | **ENVISION** | **CEEQUAL** |
|---|---|---|---|
| ENVIRONMENTAL |  | 44.86% | 42.60% |
| SOCIAL |  | 32.33% | 24.11% |
| GOVERNANCE |  | 22.81% | 33.30% |
|  | Total | 100.00% | 100.00% |

Table 10 shows the percentage distribution of credits of the certification systems in each of the thematic areas of the safeguards, detailed in each financial institution previously selected.

**Table 10.** High-level matrix total scores. Own elaboration.

|  |  | **Integration of Safeguard System Credits** | | | |
|---|---|---|---|---|---|
| **ENVISION** |  | **ADB** | **IDB** | **EBRD** | **AFDB** |
| QUALITY OF LIFE |  | 41.50 | 28.00 | 45.00 | 31.00 |
| LEADERSHIP |  | 33.00 | 21.50 | 37.50 | 31.50 |
| RESOURCE ALLOCATION |  | 15.00 | 13.50 | 16.50 | 15.50 |
| NATURAL WORLD |  | 20.00 | 20.00 | 20.00 | 20.00 |
| CLIMATE AND RISK |  | 17.00 | 19.50 | 20.50 | 20.00 |
|  | TOTAL | 126.50 | 102.50 | 139.50 | 118.00 |
| **CEEQUAL** |  | **ADB** | **IDB** | **EBRD** | **AFDB** |
| MANAGEMENT |  | 101.50 | 79.00 | 110.00 | 95.50 |
| RESILIENCE |  | 26.00 | 29.50 | 32.50 | 32.50 |
| COMMUNITIES AND STAKEHOLDERS |  | 135.50 | 79.50 | 120.00 | 110.00 |
| LAND USE AND ECOLOGY |  | 60.00 | 54.50 | 62.50 | 57.00 |
| LANDSCAPE AND HISTORIC ENVIRONMENT |  | 68.50 | 56.50 | 82.00 | 77.50 |
| POLLUTION |  | 70.50 | 54.50 | 70.50 | 57.50 |
| RESOURCES |  | 85.50 | 78.00 | 83.00 | 83.00 |
| TRANSPORT |  | 60.00 | 24.00 | 50.00 | 43.00 |
|  | TOTAL | 607.50 | 455.50 | 610.50 | 556.00 |

Table 10 below shows the points achieved by the safeguards, per credit, for each of the ENVISON and CEEQUAL systems, as well as the final totals.

The area chart in Figure 5 below shows the degree of adoption of safeguards with respect to the credits of the two systems considered. The upper part of the figure compares the relationship of safeguards with ENVISION and the lower part with CEEQUAL.

As can be seen in Figure 5 above, the safeguard policies presented a polygonal adaptation area for ENVISION credits with different densities but with certain common results. The credit that was least integrated into all safeguards was "Climate and Risk" as it was in the range of [17;20.5] points. Credits such as "Resource Allocation" and "Natural World" were also poorly covered by the safeguards, reaching ranges of [13;15] points and [20] points, respectively. It is graphically noticeable that the credit with the highest uptake in safeguard policies was "Quality of Life", with a range of [28;45] points. Finally, the "Leadership" credit was more stably integrated by most of the organizations analyzed, with a score of [31.5;37.5], except for the IDB safeguard, which barely reached [21.5] points. By way of summary, it can be argued that the safeguard policies that were best adapted to the ENVISION system were those of ADB and EBRD, although, in some credits, they had very low scores; the IDB safeguard policy had the worst adaptability, with low points totals for all the credits of both systems.

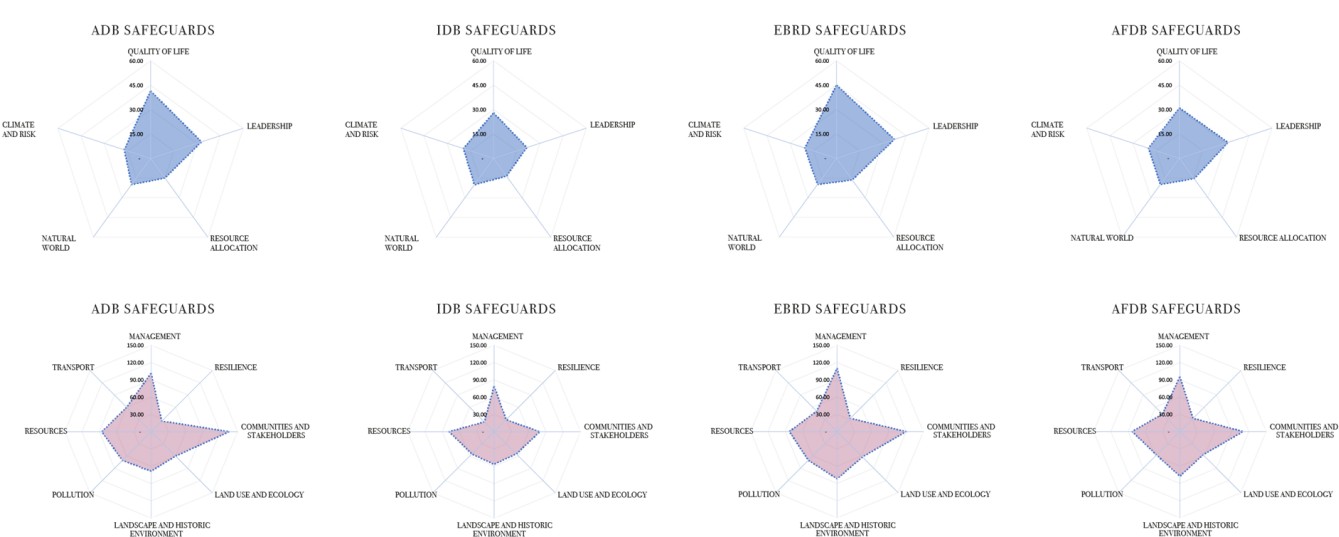

**Figure 5.** Distribution of ENVISION and CEEQUAL credits across MDB ESG safeguards. Own elaboration.

Regarding the ENVISION analysis and the results obtained, it can be argued that the EBRD safeguard led with the "Quality of Life" credit, which suggests a much stronger focus than the rest of the banks on improving quality of life through specific areas such as pollution prevention and reduction; the protection of communities within "cross-border projects", "indigenous peoples", "informed consent and processes", and "grievance and consultation mechanisms"; and in general aspects much more linked to insurance and working conditions. As for the "Leadership" credit, both the EBRD and ADB safeguards again stood out from the rest, showing strong leadership skills in their implementation, especially on issues related to "community planning and long-term monitoring", with a focus on projects throughout their life cycle, which, according to the results, was articulated in the social and governance thematic areas. From the results obtained for the "Resource Allocation" credit, all the safeguards showed very low scores due to the fact that the environmental thematic area did not include the materiality that ENVISION considers important to control, such as "Reducing water consumption during construction and operation". This is probably due to the fact that the safeguards are not very operational, as has already been mentioned.

Regarding the results assessed within the "Natural World" credit, all the safeguards also presented low and equal scores and, furthermore, they were all integrated in an integral manner only in the thematic areas with an environmental focus, achieving particular results in issues such as "land use", "soil quality", and the "control of wastewater and biodiversity". Finally, regarding the "Climate and Risk" credit, scores appeared very even across all safeguards, suggesting a very homogeneous approach among all safeguards in climate risk mitigation and that they did not adequately integrate issues such as "Climate change assessment, "Greenhouse gas reduction and assessing the vulnerability of projects with respect to their sustainable future".

In the case of the CEEQUAL system, all the graphs show a less compact area, with more peaks of variation between credits; this is because at least three of the credits studied obtained very low scores. These were "Transport", with the IDB safeguard scoring the lowest, with only 24; "Resilience"; and "Land Use and Ecology", also with the lowest score for IDB policies. The three credits "Resources", "Management", and "Communities and Stakeholders" scored high in all the safeguards analyzed, with ADB scoring the highest, with a difference of [63.5] points. The "Pollution and Landscape" and "Historic Environment" credits had high score ranges, with [54.5;70] and [56.5;82], respectively. In summary, it can be said that the safeguard policies that were best adapted to the CEEQUAL system were those of ADB and EBRD, although, for some credits, they had very low scores,

and the IDB safeguard policy had the worst adaptability, with low points totals for all the credits, far below the other three safeguard policies.

In the case of the CEEQUAL system, all the graphs show a less compact area, with more peaks of variation between credits; this is because at least three of the credits studied obtained very low scores. These were "Transport", with the IDB safeguard scoring the lowest, with only 24; "Resilience"; and "Land Use and Ecology", also with the lowest score for IDB policies. The three credits "Resources", "Management", and "Communities and Stakeholders" scored high in all the safeguards analyzed, with ADB scoring the highest, with a difference of [63.5] points. The "Pollution and Landscape" and "Historic Environment" credits had high score ranges, with [54.5;70] and [56.5;82], respectively. In summary, it can be said that the safeguard policies that were best adapted to the CEEQUAL system were those of ADB and EBRD, although, for some credits, they had very low scores, and the IDB safeguard policy had the worst adaptability, with low points totals for all the credits, far below the other three safeguard policies.

With respect to the CEEQUAL analysis and the results obtained, for the "Management" credit, the EBRD and ADB safeguards showed high scores, suggesting strong management in their safeguards in areas such as "Environmental Management" and "Sustainability Leadership", which were highly integrated into the specific environmental and governance areas of the safeguards.

For the "Resilience" credit, the scores of the different safeguards were the most even and the lowest of all the credits analyzed. The EBRD and AFDB safeguards led slightly, indicating that these two banks value resilience in a similar way by integrating issues such as "Risk assessment and mitigation" and "Floods and surface water runoff" specifically in the environmental thematic area. The "Communities and Stakeholders" credit was one of the most integrated across the safeguards and here the ABS safeguard in particular led, showing a strong focus on the inclusion of specific issues such as "Consultation and Engagement" and "Broader Social and Economic Benefits", integrating into social areas on issues such as "Indigenous Peoples", "Free, Prior and Informed Consent by Indigenous Peoples", "Economic Displacement (Including Loss of Resource/Land Use)".

Accordingly, the specific credits of "Land Use and Ecology" and "Landscape and Historic Environment" scores varied, but with EBRD safeguards leading slightly. This is because although all safeguards had scores for "Land Use and Land Value", "Land Pollution and Land Use", "Land Use and Ecology", "Landscape and Historic Environment", "Land Contamination and Land Reclamation", and "Biodiversity Protection", but the EBRD safeguard integrated many more issues, such as "Biodiversity Change and Enhancement". Regarding the scores obtained for "Heritage", all safeguards had a very good consideration of these elements, and specific documents for the control of this area were integrated into all of them. The "Pollution" credit was an important issue that all safeguards incorporated in their environmental thematic areas such as "Pollution Prevention and Abatement", with high scores for elements such as "Water and soil pollution" and "Noise and landscape pollution".

Regarding the credit "Resources", the scores were high and even, integrating mainly in themes such as "Environmental Impact of Construction Products" and "Construction Waste Management", integrated into the environmental and governance thematic areas, specifically in the thematic areas of "Resource Efficiency" and "Hazardous waste/materials, including pesticide management". To finish, the credit "Transport" appeared with inconsistent scores across safeguards and was most covered by ADB safeguards followed by the EBDR. Topics such as "Construction logistics", "Minimizing construction traffic disruptions", "Movement of construction materials", and "Movement of construction materials" were poorly incorporated in all the safeguards and only appeared in the thematic area of "Governance" in the theme of "Environmental assessment", as reflected in the scores achieved. This can be interpreted as a safeguards approach that was not aligned with practices in this area during project construction.

In conclusion and as discussed above, the safeguards policies of multilateral banks and sustainability certification schemes such as CEEQUAL and ENVISION have related purposes but are applied in different ways and this is reflected in the results obtained from the high-level matrix assessment.

*5.2. Qualitative Analysis Results*

As defined in the previous section, the tool used to carry out the qualitative analysis showed the degree of co-occurrence between the thematic areas that managed the safeguard policies and the credits that integrated the two selected sustainability certification tools. The degree of integration of the selected indicators defining these tools in the safeguards was therefore obtained in detail.

5.2.1. ENVISION Co-Occurrences

The results obtained in the case of ENVISION, highlighted in the following Table 11, show that in practically all the safeguards, the most integrated credit was "Climate and Resilience", with a single exception, that of the ABS safeguard policy, where the highest was "Quality of Life", with an average value of [0.5225]. With respect to the average number of co-occurrences, the lowest was the "Natural Word" credit, with [0.099]. None of the safeguards presented co-occurrences higher than the average in all the credits; only the ABS and EBRD policies were higher in four of the five credits, this not being fulfilled for the "Natural World" credit for EBRD or "Resource Allocation" for ABS. The lowest co-occurrence was found for the "Quality of Life" credit in the IDB policy, for which a score of [0.022] was reached—almost a 0.080 difference with respect to the average achieved by the four safeguards. On the other hand, the highest co-occurrence score was achieved by the ABS safeguard in the "Climate and Resilience" credit, with [0.7396], almost 0.22 above the average.

**Table 11.** Results of co-occurrence for ENVISION. Own elaboration.

| | | Climate and Resilience Gr = 23 | Leadership Gr = 49 | Natural World Gr = 14 | Quality of Life Gr = 17 | Resource Allocation Gr = 9 | Totals |
|---|---|---|---|---|---|---|---|
| | | Coefficient | Coefficient | Coefficient | Coefficient | Coefficient | |
| ADB | Environmental Gr = 30 | 0.7097 | 0.0260 | 0.1000 | 0.1750 | 0.1143 | 1.124937 |
| | Governance Gr = 54 | 0.0267 | 0.1075 | 0.0149 | 0.0290 | 0.0000 | 0.178105 |
| | Social Gr = 284 | 0.0033 | 0.0605 | 0.0068 | 0.0203 | 0.0000 | 0.090874 |
| | Totals | 0.7396 | 0.1940 | 0.1217 | 0.2243 | 0.1143 | |
| | | Climate and Resilience Gr = 16 | Leadership Gr = 91 | Natural World Gr = 20 | Quality of Life Gr = 11 | Resource Allocation Gr = 21 | Totals |
| | | Coefficient | Coefficient | Coefficient | Coefficient | Coefficient | |
| IDB | Environmental Gr = 41 | 0.2955 | 0.0313 | 0.1509 | 0.0196 | 0.1273 | 0.624529 |
| | Governance Gr = 6 | 0.0000 | 0.0319 | 0.0000 | 0.0000 | 0.0385 | 0.070377 |
| | Social Gr = 723 | 0.0027 | 0.0436 | 0.0054 | 0.0027 | 0.0040 | 0.058498 |
| | Totals | 0.2982 | 0.1068 | 0.1564 | 0.0223 | 0.1698 | |

**Table 11.** *Cont.*

| | | Climate and Resilience Gr = 39 | Leadership Gr = 96 | Natural World Gr = 13 | Quality of Life Gr = 23 | Resource Allocation Gr = 22 | Totals |
|---|---|---|---|---|---|---|---|
| | | Coefficient | Coefficient | Coefficient | Coefficient | Coefficient | |
| EBRD | Environmental Gr = 62 | 0.6290 | 0.0128 | 0.0563 | 0.0241 | 0.2727 | 0.995014 |
| | Governance Gr = 29 | 0.0000 | 0.1062 | 0.0000 | 0.0196 | 0.0000 | 0.125803 |
| | Social Gr = 204 | 0.0167 | 0.0638 | 0.0000 | 0.0657 | 0.0000 | 0.146294 |
| | Totals | 0.6458 | 0.1828 | 0.0563 | 0.1094 | 0.2727 | |
| | | Climate and Resilience Gr = 31 | Leadership Gr = 34 | Natural World Gr = 16 | Quality of Life Gr = 15 | Resource Allocation Gr = 24 | Totals |
| | | Coefficient | Coefficient | Coefficient | Coefficient | Coefficient | |
| AFDB | Environmental Gr = 72 | 0.3919 | 0.0095 | 0.0602 | 0.0357 | 0.1707 | 0.668103 |
| | Governance Gr = 14 | 0.0000 | 0.0909 | 0.0000 | 0.0000 | 0.0000 | 0.090909 |
| | Social Gr = 177 | 0.0146 | 0.0657 | 0.0052 | 0.0159 | 0.0101 | 0.111422 |
| | Totals | 0.4065 | 0.1661 | 0.0654 | 0.0516 | 0.1808 | |

Finally, the results of total co-occurrences for each safeguard were obtained, as can be seen in the following Table 12. The highest co-occurrence was obtained for ADB and EBRD, with a difference of [0.64] between them, and with respect to those of IDB and AfDB, reaching a difference with respect to their totals of [0.09].

**Table 12.** Summary of co-occurrence scores of ENVISION with MDB ESG safeguard areas. Own elaboration.

| | ADB | IDB | EBRD | AFDB | Average |
|---|---|---|---|---|---|
| ENVIRONMENT | 1.1249 | 0.625 | 0.9950 | 0.6681 | 0.85315 |
| GOVERNANCE | 0.1781 | 0.070 | 0.1258 | 0.0909 | 0.11630 |
| SOCIAL | 0.0909 | 0.058 | 0.1463 | 0.1114 | 0.10177 |
| Totals | 1.3939 | 0.7534 | 1.2671 | 0.8704 | |

In the following Table 13, we observe the same results presented in Table 12 but based on ENVISION credits.

**Table 13.** Summary of co-occurrence scores of MDB ESG safeguard areas with ENVISON credits. Own elaboration.

| | ADB | IDB | EBRD | AFDB | Average |
|---|---|---|---|---|---|
| CLIMATE AND RESILIENCE | 0.7396 | 0.2982 | 0.6458 | 0.4065 | 0.5225 |
| LEADERSHIP | 0.1940 | 0.1068 | 0.1828 | 0.1661 | 0.1624 |
| NATURAL WORLD | 0.1217 | 0.1564 | 0.0563 | 0.0654 | 0.1000 |
| QUALITY OF LIFE | 0.2243 | 0.0223 | 0.1094 | 0.0516 | 0.1019 |
| RESOURCE ALLOCATION | 0.1143 | 0.1698 | 0.2727 | 0.1808 | 0.1844 |
| Totals | 1.3939 | 0.7534 | 1.2671 | 0.8704 | |

As we had highlighted previously the visual representation provided by the heatmap serves as an analytical tool, offering an immediate, intuitive grasp of the comparative performance of the institutions under consideration—namely ADB, IDB, EBRD, and AFDB—across a spectrum of environmental and governance parameters. This visual format utilizes a color gradient to symbolize the range of scores, with darker hues corresponding to higher scores and lighter ones to lower scores, creating a visually stratified assessment.

The Sankey graphs included in Figure 6 clearly exhibit that the thematic areas most connected with the ENVISION credits were the environmental and social areas, with the area associated with governance or processes being far behind. This last area was integrated in very few credits: only in the ADB safeguard policy, for which it was integrated with very low co-occurrences in four of the five credits, with "Resource Allocation" not covered. In the case of EBRD, it was found only in two credits, "Leadership" and "Quality of Life", and, finally, for IDB and AfDB, it was only found for the "Leadership" credit, with a very low co-occurrence.

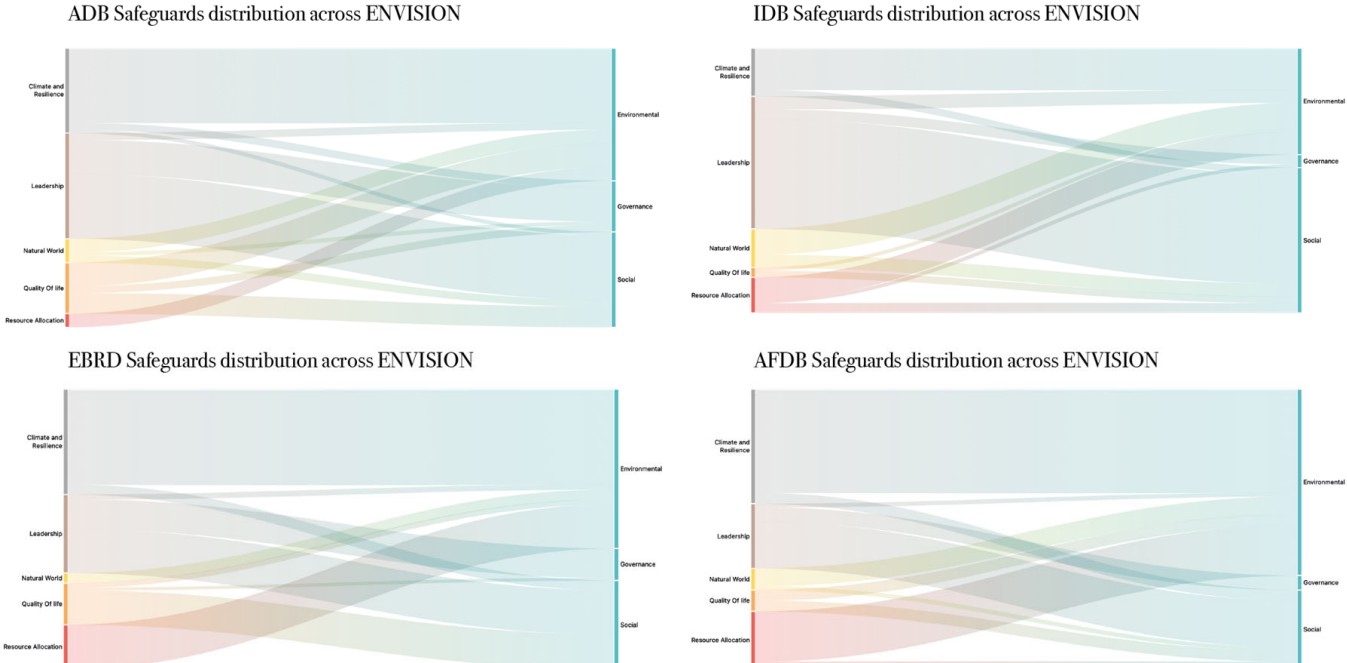

**Figure 6.** Sankey charts with distribution of ENVISION credits in thematic areas of safeguards according to qualitative analysis.

To recap and summarize in the case of ENVISION, integrating the areas contained in the parameters included in the "Quality of Life" credit would provide projects with more insight into aspects that, a priori, seem reasonable for the safeguards to include. Examples are addressing the impact on the health of the communities and inhabitants of the communities affected by or hosting the infrastructure as well as aspects important to control during the execution of projects, such as the physical safety of residents and workers and the mitigation of impacts and nuisances during construction. In addition, the low adoption of these credits in the scope of the safeguards analyzed can affect the functionality of the infrastructure, something that undoubtedly impacts the environment and above all the sensitivity of the community affected by the infrastructure. Finally, the low adoption of the parameters defined by the "Natural World" credit, such as the biodiversity of the site, as well as soil and water, considerably affects the conservation of areas with high diversity or ecological value and the safeguarding of surface water regimes and aquifers that may be altered during the construction and operation of an asset. The incorporation of these parameters within safeguard policies is of high importance, as incorporating designs that

consider these control points will allow for very high sustainability values to be achieved at the end of an infrastructure's life cycle.

### 5.2.2. CEEQUAL Co-Occurrences

The results obtained in the case of CEEQUAL in Table 14 show that in practically all the safeguards, the credit that was most integrated was that of "Communities and Stakeholders", with an average value of 0.6403. The lowest average number of co-occurrences was for "Transport" with 0.0010, where only the IDB safeguard integrated co-occurrences, and no co-occurrences were found in the rest of the safeguards.

**Table 14.** Results of co-occurrence for CEEQUAL. Own elaboration.

| ADB | | Management Gr = 31 | Resilience Gr = 29 | Communities and Stakeholders Gr = 319 | Land Use and Ecology Gr = 70 | Landscape and Historic Environment Gr = 14 | Pollution Gr = 1 | Resources Gr = 8 | Transport Gr = 1 | Totals |
|---|---|---|---|---|---|---|---|---|---|---|
| | | Coefficient | Coefficient | Coefficient | Coefficient | Coefficient | Coefficient | Coefficient | Coefficient | |
| | Environment Gr = 30 | 0.0339 | 0.0351 | 0.0325 | 0.1765 | 0.0233 | 0.0333 | 0.1176 | 0.0000 | 0.419 |
| | Governance Gr = 72 | 0.3733 | 0.0412 | 0.1171 | 0.0441 | 0.0118 | 0.0000 | 0.0127 | 0.0000 | 0.600 |
| | Social Gr = 284 | 0.0535 | 0.0330 | 0.6210 | 0.1028 | 0.0171 | 0.0000 | 0.0034 | 0.0000 | 0.831 |
| | Totals | 0.4607 | 0.1093 | 0.7707 | 0.3234 | 0.0521 | 0.0333 | 0.1337 | 0.0000 | |

| IDB | | Management Gr = 107 | Resilience Gr = 186 | Communities and Stakeholders Gr = 424 | Land Use and Ecology Gr = 20 | Landscape and Historic Environment Gr = 18 | Pollution Gr = 2 | Resources Gr = 12 | Transport Gr = 3 | Totals |
|---|---|---|---|---|---|---|---|---|---|---|
| | | Coefficient | Coefficient | Coefficient | Coefficient | Coefficient | Coefficient | Coefficient | Coefficient | |
| | Environment Gr = 41 | 0.0137 | 0.0318 | 0.0109 | 0.0517 | 0.0172 | 0.0488 | 0.1277 | 0.0000 | 0.302 |
| | Governance Gr = 6 | 0.0089 | 0.0000 | 0.0047 | 0.0000 | 0.0000 | 0.0000 | 0.0000 | 0.0000 | 0.014 |
| | Social Gr = 723 | 0.0285 | 0.0168 | 0.5232 | 0.0068 | 0.0109 | 0.0000 | 0.0068 | 0.0041 | 0.597 |
| | Totals | 0.0511 | 0.0486 | 0.5388 | 0.0585 | 0.0282 | 0.0488 | 0.1345 | 0.0041 | |

| EBRD | | Management Gr = 70 | Resilience Gr = 43 | Communities and Stakeholders Gr = 340 | Land use and ecology Gr = 60 | Landscape and historic environment Gr = 39 | Pollution Gr = 8 | Resources Gr = 17 | Transport Gr = 6 | Totals |
|---|---|---|---|---|---|---|---|---|---|---|
| | | Coefficient | Coefficient | Coefficient | Coefficient | Coefficient | Coefficient | Coefficient | Coefficient | |
| | Environment Gr = 62 | 0.0476 | 0.0606 | 0.0361 | 0.1193 | 0.0000 | 0.0938 | 0.1970 | 0.0000 | 0.554 |
| | Governance Gr = 30 | 0.4286 | 0.0282 | 0.0571 | 0.0227 | 0.0000 | 0.0000 | 0.0000 | 0.0000 | 0.537 |
| | Social Gr = 204 | 0.0787 | 0.0466 | 0.5111 | 0.0602 | 0.0385 | 0.0000 | 0.0000 | 0.0000 | 0.735 |
| | Totals | 0.5549 | 0.1354 | 0.6043 | 0.2022 | 0.0385 | 0.0938 | 0.1970 | 0.0000 | |

| AFDB | | Management Gr = 65 | Resilience Gr = 27 | Communities and Stakeholders Gr = 264 | Land Use and Ecology Gr = 40 | Landscape and Historic Environment Gr = 25 | Pollution Gr = 14 | Resources Gr = 17 | Transport Gr = 0 | Totals |
|---|---|---|---|---|---|---|---|---|---|---|
| | | Coefficient | Coefficient | Coefficient | Coefficient | Coefficient | Coefficient | Coefficient | Coefficient | |
| | Environment Gr = 72 | 0.0620 | 0.0421 | 0.0633 | 0.1200 | 0.0104 | 0.0886 | 0.2027 | 0.0000 | 0.589 |
| | Governance Gr = 14 | 0.0533 | 0.0789 | 0.0258 | 0.0000 | 0.0000 | 0.0000 | 0.0000 | 0.0000 | 0.158 |
| | Social Gr = 177 | 0.0568 | 0.0515 | 0.5583 | 0.0284 | 0.0151 | 0.0160 | 0.0052 | 0.0000 | 0.731 |
| | Totals | 0.1721 | 0.1726 | 0.6474 | 0.1484 | 0.0255 | 0.1046 | 0.2079 | 0.0000 | |

None of the safeguards showed higher than average co-occurrences in all credits; only the EBRD policies were higher in six of the eight credits, "Communities and Stakeholders" and "Transport" not being integrated. The lowest co-occurrence was found for the "Landscape and Historic Environment" credit in the AfDB policy, where a score of 0.025 was achieved, but, in any case, it was a very close result to the average achieved by the rest

of the safeguard policies, which was 0.036. On the other hand, the highest co-occurrence score was achieved by the ABS safeguard in the "Communities and Stakeholders" credit, with 0.74066 with a 0.13 difference with respect to the average.

Finally, the results of total co-occurrences for each safeguard were obtained, as can be seen in Table 15, highlighting that the highest co-occurrence was obtained for ADB and EBRD, with a difference of [0.06] between them.

**Table 15.** Summary of co-occurrence scores of CEEQUAL with MDB ESG safeguard areas. Own elaboration.

|  | ADB | IDB | EBRD | AFDB | Average |
|---|---|---|---|---|---|
| ENVIRONMENT | 0.4189 | 0.3018 | 0.5543 | 0.5891 | 0.4660 |
| GOVERNANCE | 0.6003 | 0.0136 | 0.5366 | 0.1581 | 0.3271 |
| SOCIAL | 0.8308 | 0.5972 | 0.7352 | 0.7313 | 0.7236 |
| Totals | 1.8499 | 0.9126 | 1.8261 | 1.4785 |  |

The colour coding of Heat map highlighted on Tables 15 and 16 with the same approach and meaning as mentioned above.

**Table 16.** Summary of co-occurrence scores MDB ESG safeguard areas with ENVISON credits. Own elaboration.

|  | ADB | IDB | EBRD | AFDB | Average |
|---|---|---|---|---|---|
| MANAGEMENT | 0.4607 | 0.0511 | 0.5549 | 0.1721 | 0.3097 |
| RESILIENCE | 0.1093 | 0.0486 | 0.1354 | 0.1726 | 0.1165 |
| COMMUNITIES AND STAKEHOLDERS | 0.7707 | 0.5388 | 0.6043 | 0.6474 | 0.6403 |
| LAND USE AND ECOLOGY | 0.3234 | 0.0585 | 0.2022 | 0.1484 | 0.1831 |
| LANDSCAPE AND HISTORIC ENVIRONMENT | 0.0521 | 0.0282 | 0.0385 | 0.0255 | 0.0360 |
| POLLUTION | 0.0333 | 0.0488 | 0.0938 | 0.1046 | 0.0701 |
| RESOURCES | 0.1337 | 0.1345 | 0.1970 | 0.2079 | 0.1683 |
| TRANSPORT | 0.0000 | 0.0041 | 0.0000 | 0.0000 | 0.0010 |
| Totals | 1.8833 | 0.9126 | 1.8261 | 1.4785 |  |

In the following Table 16, we observe the same results presented in Table 15 but based on CEEQUAL credits.

The Sankey graphs illustrated in Figure 7 show that the thematic areas most connected with CEEQUAL credits were the environmental and social areas, with the area associated with governance or processes being far behind. This last area was integrated in very few credits: only in the safeguard policies of ADB and EBRD, being integrated with co-occurrences in six of the eight credits for ADB and in four of eight for EBRD. The credits not covered in the case of ADB were "Pollution" and "Transport" and in the case of EBRD were "Landscape and Historic Environment", "Pollution", "Resources", and Transport.

To recap and summarize, in the case of the CEEQUAL system, the adoption of "Landscape and Historic Environment", "Pollution", and "Transport" credits in the safeguards analyzed was very low, even in the case of the "Transport" credit. While it is true that the "Transport" area considers the effective management of the impacts of all modes of transport, both during construction and during operation, it also includes the movement of construction materials, the waste generated, and the transport of the construction workforce itself, as well as the disruption of the affected population and users of the transport network during the life of the asset. Therefore, we consider that this is an area that the safeguards should contain within their scope. Regarding the "Landscape and Historic Environment" area, which analyzes the parameters that contribute to the conservation of the landscape and associated heritage elements in and around the project site, the integration of these

parameters in the safeguards analyzed was very low. Such integration will contribute to protecting and enhancing both landscape characteristics and heritage assets where present. This is the case not only from the point of view of the aesthetic value or visual impact of infrastructure in the surrounding environment but also the measures taken to protect and enhance the historic environment for the benefit of present and future generations of the community affected by it. Perhaps most relevant is the low value achieved in the "Pollution credit", the third lowest value achieved in the qualitative analysis. All safeguards analyzed had areas pertaining to pollution prevention and reduction, such as the control of hazardous waste or materials, including the handling of pesticides, but did not contain parameters that allowed for the control of air, water, and noise pollution resulting from the construction and future operation of assets. With such measures, we believe that the approach proposed within the safeguards for pollution control, undoubtedly an important sustainability parameter to control, could be greatly improved.

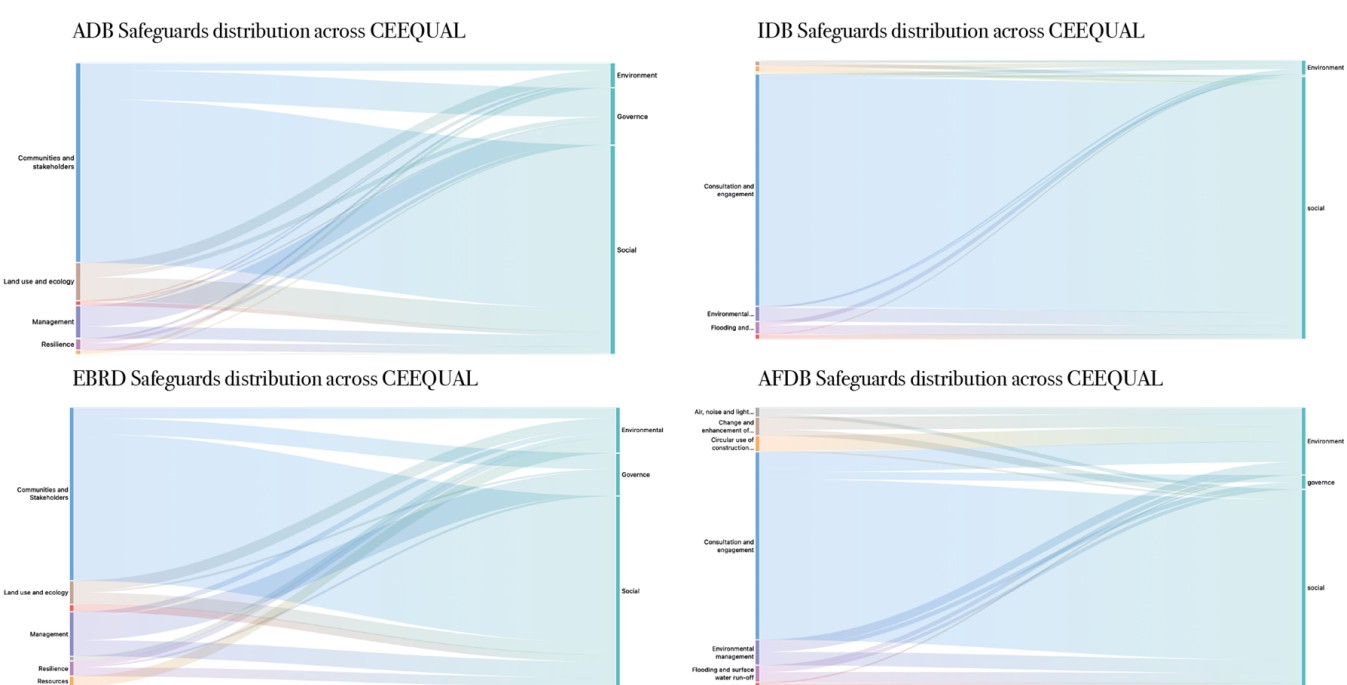

**Figure 7.** Sankey charts with distribution of CEEQUAL credits in safeguards thematic areas according to qualitative analysis.

## 6. Conclusions

As we have already said, it should be understood that the ESG safeguards policies employed by MDBs seek to mitigate risks and promote sustainable practices in large-scale projects, generally encompassing high-level concepts of social and environmental issues and governance, while the CEEQUAL and ENVISION schemes promote operational and technical excellence at the project level. This has been a major challenge that has posed several important methodological constraints. Firstly, there was heterogeneity in the sustainability metrics, given that the credits or thematic areas of assessment proposed in the selected ENVISION and CEEQUAL systems and those used by MDBs' ESG safeguards policies had to be aligned, which led to inconsistencies in the interpretation and design of the methodologies used so that the methodologies used resorted to qualitative assessment. Secondly, the identification and selection of codes, sub-codes, and indicators for both certification systems and safeguards had to conform to criteria of applicability and relevance, and, in certain CEEQUAL credits, it was difficult to standardize them. Another challenge we faced was to compile and group all the information from the banks' safeguards, which, in many cases, meant eight or even nine different documents needed to be processed in an integrated way with the software used.



This paper presented a comparative analysis of the extent to which safeguard policies employed by multilateral infrastructure development organizations integrate the sustainability measurement credits or checkpoints proposed in common infrastructure sustainability assessment tools. The aim was to contribute to the overview and debate on whether these safeguard policies cover all aspects of sustainability necessary to comply with international certifications. The proposed comparative framework is mainly based on a combination of two tools with the same approach, consisting of the comparison of ESG areas and the ENVISION and CEEQUAL credits.

Do MDB safeguard policies comply with all the items and indicators monitored in the credits of the international sustainability certificates?

The answer confirmed by the analyzed data (See Supplementary Materials) is that they do not. Both in the results obtained in the high-level matrix and confirmed by the qualitative analysis carried out, there is a clear lack of integration in governance, which can be seen in the very low scores obtained for the credits of the two systems analyzed.

Furthermore, there is a contradiction in the future development of safeguard policies ensuring that all areas of sustainability are guaranteed. On the one hand, there is a demand for approaches to have a more detailed assessment performance tailored to local market sensitivities and infrastructure typology, which means, among other things, that they should be more case- and location-specific. At the same time, there is a demand for these safeguards to be broader so that they are more adaptable to a wide group of users for different case circumstances. There is also a need for these tools to have better-integrated and more standardized areas and to be able to provide more transparent results. In line with some points highlighted by Humphrey [23], we agree that the ESG safeguard approach used by the major multilateral development banks requires a thorough rethink to address conceptual and practical shortcomings.

Can the future development of new safeguard policies respond to the challenges of providing benchmarks and performance indicators, better guidelines, and greater availability of implementation data, and thus be utilized to conduct ex-post analyses of the impacts on infrastructure sustainability? As with the multiple facets of the sustainability concept itself, the development of more comprehensive and integrated safeguards can only happen when all parameters that have an impact during the design and construction phase of the asset are considered, something that is not the case today.

Efforts have been made by multilateral organizations to implement safeguard policies in terms of tools to control the sustainability of the projects they finance, not only for the conceptualization and design phase but also for the subsequent construction phase, as well as emphasizing the care and commitment they require from borrowers to ensure compliance. Despite this, the analysis shows that the tools are limited in scope and do not incorporate important aspects in achieving guarantees for the sustainability of infrastructure. The literature review conducted in this work shows that, although the concept of sustainability has become increasingly important, the ex-post assessment of the sustainability of infrastructure projects remains an unresolved issue, and this is demonstrated even more so if we try to integrate it using international certificate credits and tools. This does not mean that the MDB safeguards are a complete failure—far from it. The safeguards have had a very positive impact, improving the way MDBs design and implement projects, reducing negative social and environmental impacts and repairing them when they occur [17].

The two ENVISION CEEQUAL certificate tools analyzed are biased toward environmental approaches, to the detriment of the more economic and social dimensions [26], the latter being a top priority for multilateral safeguards as well as for developing countries that host the infrastructure financed by these entities and where the promotion of economic growth and sustainable living is a predominant objective when making an investment. This is perhaps the first point to consider in conclusion, as the integration of many of the parameters included in the environmental credits of the two ENVISION and CEEQUAL certificates was not included in the safeguards analyzed, given that the design of these safeguards was mostly not based on the existing legal and regulatory frameworks in the countries

where the projects were located and that they were not able to provide comprehensive environmental requirements and guidelines for the projects.

Consequently, although the current safeguard approaches enable parameters that affect the sustainability of infrastructure to be controlled, none of them can be used to carry out a holistic assessment. However, we note that the considerable work achieved through public policy itself, as well as the accumulated knowledge and experience of the staff managing these ESG safeguards at multilateral development banks, would be extremely valuable in convincing governments of the benefits of a more holistic and comprehensive approach that considers many of the parameters not currently included from the outset. Such an approach could also aid in providing practical assistance and best practice examples to help strengthen the monitoring frameworks of countries hosting infrastructure. Finding a tool that encompasses all the features of international certification schemes for assessing the sustainability of projects would be a complicated issue due to the great difference between the regions or countries in which they would be applied as well as the nature of the projects themselves and the necessary compromise between accuracy and feasibility. This could be achieved by integrating the different documents into a single one, consolidating and standardizing the same areas of action, which could well be the same ESG areas, as well as all the parameters that can be considered in infrastructure projects, or by integrating clear and well-defined indicators, as well as their measurement and comparison methodologies, into the safeguard policies themselves.

Finally, the taxonomy proposed by the EU and the growing environmental pressures in decision-making for the financing of infrastructure projects associated with environmental protection and the prevention of the breakdown of economic and social systems can be considered a perfect impetus for new lines of research that seek to analyze how this taxonomy could be integrated into the current ESG safeguard policies of these multilateral organizations.

To enhance the alignment between public authorities, multilateral banks, and companies holding sustainability certifications like ENVISION or CEEQUAL, a three-phase integration approach is recommended. Initially, a universal taxonomy should be established to align diverse ESG metrics, thematic assessment areas, and indicators, streamlining them into a unified sustainability framework. This would involve the creation of a hybrid certification model that merges high-level policy safeguards with detailed project-level operational criteria, tailored to local contexts yet maintaining global sustainability principles. A subsequent phase would introduce a centralized digital platform for standardized data management, fostering the transparent and consistent application of qualitative and quantitative assessments across various projects. This platform would support the cross-comparison of sustainability parameters and facilitate a more rigorous post-implementation analysis of infrastructure impacts. Lastly, it is essential to refine safeguard policies to balance specificity for local market sensitivities with broad applicability for diverse user groups and scenarios. This includes incorporating overlooked ESG aspects into all project phases, particularly governance, and leveraging the expertise of MDBs to strengthen host countries' monitoring frameworks. Such integrative steps would ensure a holistic, adaptable approach to sustainability assessment, in line with the EU taxonomy, and enhance the decision-making process for sustainable infrastructure financing. This alignment could serve as a catalyst for further research and policy development, ensuring comprehensive coverage of sustainability dimensions in infrastructure projects.

**Supplementary Materials:** The following supporting information can be downloaded at: https://www.mdpi.com/article/10.3390/su16093789/s1 (data; images; figures).

**Author Contributions:** D.R.E.: Conceptualization, Methodology, Software, Data Maturity, Drafting—Preparation of the Original Draft, Visualization, Research. R.M.A.R.: Writing and Editing, Data Maturity, Visualization, Supervision, Validation, Revision. All authors have read and agreed to the published version of the manuscript.

**Funding:** This research received no external funding.

**Institutional Review Board Statement:** Not applicable.

**Informed Consent Statement:** Not applicable.

**Data Availability Statement:** The data presented in this study are available on request from the corresponding author.

**Conflicts of Interest:** The authors declare no conflicts of interest.

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
