# Peer review of "Assessing Multilateral Development Bank ESG Safeguard Integration with International Sustainability Ratings"

_sustainability, doi:10.3390/su16093789_

Round 1

Reviewer 1 Report

Comments and Suggestions for Authors

I am delighted to have the chance to review the article that aims to “find out the degree to which the ESG safeguards used by Multilateral Development Banks (MDBs) in the design and implementation phase of infrastructure projects of all types are related to the credits that two internationally recognised and prestigious international systems use to determine the degree of sustainability of infrastructure projects” (lines 244 to 247). After a thorough review, I would like to share the following comments with the authors:

1.      The title should be revised to remove the grammatical mistake and to make it more precise and concise;

2.      The authors should refer to the MDPI/Sustainability’s guidelines for authors, especially the section on writing the abstract. The abstract should be shortened and re-written to make it more precise and concise;

3.      The ESG safeguard system is focused in the article. It is important to provide the context and detailed explanations of what the ESG safeguard system is, to ensure that readers have enough context to understand the study and how the research questions were developed. Sometimes, the purpose of the article seems unclear. The aim, as quoted above, is written in lengthy English;

4.      For this reason, I would like to reconfirm the research question and research hypothesis (line 329) of the article;

5.      The use of acronyms in the article should be clarified. For example, the full names of the banks mentioned (e.g. ERDB and IDB) are not provided. Addressing this and the other comments mentioned above would greatly improve the readability of the article;

6.      The methodology, in general, is clear and readers can follow the steps. However, I suggest the authors to clarify the relationship between Figures 2 and 3 and Table 4. Besides, the process for calculating the matrix scores is unclear and more explanations should be given;

7.      Many tables, such as Tables 1, 2, 4, 6 and 12, are not linked and referenced in the text. Table 8 is missing. In Table 7, the term “NO INDICATORS” is confusing – does it mean “number of indicators” or that there are no indicators? In Figure 5, the texts are too small to read;

8.      The article appears not ready for publication due to the grammatical mistakes found, such as “Their scope is generalist” (line 309) and “I we look at …” (Line 465);

9.      The use of dots (.) and commas (,) as decimal separator is confusing in the article;

10.  At first glance, the findings seem to be inadequately discussed. The findings section only reported results without discussion. The discussions are found in the conclusions section instead. If the authors present the discussions more clearly and concisely, whether to put them with the findings or keep them in the conclusion section, readers will find it easier to follow.

Comments on the Quality of English Language

Acceptable in general. Moderate editing of English suggested before recommending for acceptance.

Author Response

  1.      The title should be revised to remove the grammatical mistake and to make it more precise and concise.

Addressed in Line 2: The paper is currently being revised by MPDI's translation services to adapt it to a more correct English but I can advance that the title could be the following "Assessing MDB´s ESG Safeguard integration with International Sustainability Ratings"

2.      The authors should refer to the MDPI/Sustainability’s guidelines for authors, especially the section on writing the abstract. The abstract should be shortened and re-written to make it more precise and concise;

Addressed and corrected in Line 12 Abstract Rewriten "

In an era where sustainability is paramount, this study critically assesses how Multilateral Development Banks (MDBs) integrate internationally recognized sustainability indicators into their ESG safeguard policies. MDB have historically incorporated policies to manage environmental and social risks in project financing, yet protections against negative impacts in developing countries often remain insufficient. On the other hand, several infrastructure sustainability rating systems have been established around the world in recent decades due to economic growth and the importance of controlling the environmental impacts associated with the construction sector. The purpose of this study is to analyze whether and how the indicators that these internationally recognized systems use to rate whether a project is sustainable are integrated into these safeguards, by using sing several methodologies as an analysis of existing documentation, a high-level matrix and qualitative methods based on co-occurrences using specialized "atlas ti" software. Results show MDBs' coverage of financial, governance, and country risks lacks the sustainability focus found in these rating systems. Therefore, the study concludes MDB´s safeguards must evolve, balancing comprehensive sustainability parameters and detailed management guidelines, and addressing impacts beyond statutory frameworks, to encourage stakeholder engagement for more sustainable infrastructure projects".

3.      The ESG safeguard system is focused in the article. It is important to provide the context and detailed explanations of what the ESG safeguard system is, to ensure that readers have enough context to understand the study and how the research questions were developed. Sometimes, the purpose of the article seems unclear. The aim, as quoted above, is written in lengthy English.

Addressed and corrected from line nº 114

4.      For this reason, I would like to reconfirm the research question and research hypothesis (line 329) of the article; 

Addressed and corrected on line nº78 with "The main objective of this study is to answer the following question  Does the MDB´s safeguard policies complies with all the items and indicators monitored in the credits of the international sustainability certificates? and to compare the safeguards policy  systems available for each selected MDB in order to assess which MDB has the highest level of sustainability areas integration proposed by the recognized infrastructure sustainability certification schemes"

5.      The use of acronyms in the article should be clarified. For example, the full names of the banks mentioned (e.g. ERDB and IDB) are not provided. Addressing this and the other comments mentioned above would greatly improve the readability of the article;

Addressed and corrected in all the paper as you can see in figure nº 1 or table nº 2

  1. The methodology, in general, is clear and readers can follow the steps. However, I suggest the authors to clarify the relationship between Figures 2 and 3 and Table 4. Besides, the process for calculating the matrix scores is unclear and more explanations should be given

Addressed and corrected on line 367 with "Figure no. 2 sets out the thematic areas and parameters that ESG safeguards seek to control in their projects and correlates directly with codes in Table no. 4, categorized by the three dimensions of sustainability  (environmental, social and governance). Each point in this table 4, such as for example "Pollution Prevention and Abatement" (MA.1) or "Indigenous Peoples" (S.1), is reflected in a specific code, providing a structured framework for systematic reference and management. In relation to this, figure 3 details the essential components in the sustainability certification schemes, which are coded in table 7, e.g. for the CEEQUAL scheme, "Flooding and Surface Water Run-Off" (2.2), or Welfare (QL1). for ENVISION. The relationship of these 2 figures and the tables is to compose a system of codes classified by area and parameters that allow us to assess their interrelation both in the high-level matrix and in the qualitative analysis through co-occurrences."

As well, it has been included the following paragraph to enrich the High level matrix points assessment  , from line 486 "The high-level matrix evaluation team assigned scores to a total of 219 indicators, broken down into 67 for ENVISION (spread over the 5 codes and 14 sub-codes) and 152 for CEEQUAL (spread over the 8 codes and 29 sub-codes) and defined in table 7, applying the following scoring scheme:

  • Fully Covered (1 point): 1 point was assigned to the indicators of the certification systems that are fully aligned with the objectives and approaches of the subject areas that integrate the different safeguards policies defined in Figure 2. An indicator with this score indicates full and aligned integration with the safeguard.
  • Partially Covered (0.5 points): 0.5 points were assigned to indicators that meet some, but not all, of the objectives of the thematic area. The indicator in this category indicates that, although partly recognized and addressed, there is still room for more comprehensive coverage.
  • Not Covered (0 points): This is assigned to indicators without any focus or content in the safeguards thematic areas. An indicator with a score of zero highlights opportunities to incorporate new elements in future revisions of certification systems"

7. Many tables, such as Tables 1, 2, 4, 6 and 12, are not linked and referenced in the text. Table 8 is missing. In Table 7, the term “NO INDICATORS” is confusing – does it mean “number of indicators” or that there are no indicators? In Figure 5, the texts are too small to read;

Addressed and corrected all figures and tables

8. The article appears not ready for publication due to the grammatical mistakes found, such as “Their scope is generalist” (line 309) and “I we look at …” (Line 465);

Addressed and corrected with professional revision by mdpi services

9. The use of dots (.) and commas (,) as decimal separator is confusing in the article;

Addressed and corrected with professional revision by mdpi services

10. At first glance, the findings seem to be inadequately discussed. The findings section only reported results without discussion. The discussions are found in the conclusions section instead. If the authors present the discussions more clearly and concisely, whether to put them with the findings or keep them in the conclusion section, readers will find it easier to follow.

The discussion has been addressed and corrected, integrating the discussion in the results part and removing those discussions from the conclusions part. It has also been rewritten.

Lines 604, 647,746,805,833

Reviewer 2 Report

Comments and Suggestions for Authors

Dear Esteemed Author(s),

Please find attached the Review Report covering several suggestions and recommendations.

Comments on the Quality of English Language

The English language should be improved.

Author Response

  • The title of the paper should be shortened. As well, the title should be adjusted in order to be more compelling. Addressed and corrected, line  1
  • The introductory section does not emphasize the novelty and originality of the analysis. As well, the author(s) should highlight the gap in the related literature.

Addressed and corrected, line  62 "It means an opportunity to carving a new niche in the academic landscape. It departs from the existing literature that mainly focuses on the role of Multilateral Development Bank (MDBs) (Environmental, Social, and Governance “ESG”) safeguard policies in advancing project sustainability, as well as on the use of ENVISION and CEEQUAL for tracking sustainable progress in ongoing projects. Our investigation explores uncharted territory. This paper's novel analysis of the integration of credits and monitoring indicators from these international certifications into MDB safeguard policies presents a significant learning opportunity for the academic community and policymakers. By dissecting the extent and manner of this integration, the study lays the groundwork for refining these policies toward more effective and actionable strategies in the construction phases of projects. Ultimately, it ensures that projects not only adhere to best practices but are also directed by a robust framework equipped with clear and measurable indicators, marking a step toward operational excellence and enhanced sustainability outcomes for projects financed by these banks."

  • The second section dedicated to prior literature should be enriched with a distinct summarizing the most recent studies in the field.

Addressed and corrected, line 114,134,209

  • The third section towards methodology is too short and should be extended to deeply discuss the employed methods.

Addressed and corrected, line 367,486

  • The analysis performed in the fourth section should be expanded with more argument regarding rating tools section and codes.

Addressed and corrected, line 367,486

  • The discussion of the results in the fifth section should also cover comparisons with earlier studies. 

 "As it has been highlighted there are no previous publications that analyze what has been discussed in this paper. Of course,  there are publications that analyze how MDB´ through their ESG Safeguards policies contribute to achieving sustainability of projects and how ENVISION and CEEQUAL contribute to monitoring progress on sustainability of already implemented projects as it has been highlighted on cap 2. Literature review, but there is none that analyses what is proposed in this paper, and that is why there is such a gap. By analyzing how and how many of these issues through the indicators used by these international certificates are already integrated into safeguards policies, it will be much easier to move towards revising these policies and orienting them towards more operational and impactful approaches to projects during construction, ensuring that projects are not only subject to good practice but also to a structured framework with clear indicators".

  • The last section should point out the research limitations.

Addressed and corrected, line  833

Reviewer 3 Report

Comments and Suggestions for Authors

The article  analyze  internationally recognized indicators used to rate whether a project is sustainable are integrated into the safeguards by multilateral organizations. The article focuses in some rating tools.

In my opinion, the article is very well written. The comparative analysis is very rigorous. I think it is publishable but in terms of information on this topic it is somewhat outdated, it does not include the legislation that has been generated by the European Commission in recent years. The bibliography is also a bit outdated, it should be reviewed with recent articles.

Acronyms should be defined, for example names of MDBs.

Author Response

I think it is publishable but in terms of information on this topic it is somewhat outdated, it does not include the legislation that has been generated by the European Commission in recent years. Comments addressed on lines 134

The bibliography is also a bit outdated, it should be reviewed with recent articles.Comments addressed on lines 209

Reviewer 4 Report

Comments and Suggestions for Authors

Abstract

The abstract suggests broad conclusions regarding the integration of sustainability indicators into ESG safeguards without specifying the extent of variability or inconsistency among different banks. 

Introduction

1- The introduction is well written, but it needs further expansion, and it -also needs to clarify the study’s contribution to the growing literature, as it is not enough to mention the objectives of the study and consider it as a contribution.

2-Furthermore, authors should conclude the introduction with a small paragraph outlining the remaining sections of the study

Literature Review

1- The review focuses on ESG safeguards and sustainability rating systems but could be enriched by integrating findings from broader literature on sustainable finance, development outcomes, or ESG impact assessment methodologies. Therefore, it is suggested that the authors incorporate more recent literature into the literature review to develop it further, such as:

 https://doi.org/10.1002/csr.2563

2-The review summarizes various sources, it could further critique the methodologies, biases, or limitations of the existing research. This would strengthen the case for the study's contribution to filling knowledge gaps.

3- There is an absence of a theoretical framework

Methodology

The rationale for choosing the specific methodologies (documentation analysis, high-level matrix, qualitative methods) could be more thoroughly justified in terms of how they best address the research questions.

Analysis

The discussion of the analysis of the results is limited to their narration only, without any possible contribution from the authors. The authors must develop and improve the discussion based on the empirical and theoretical literature.

Conclusion

 1-The conclusion could be strengthened by providing more concrete, actionable recommendations for MDBs, policymakers, or sustainability rating organizations on how to improve the integration of ESG safeguards.

2-Suggestions for future research could be more specific, perhaps pointing to particular aspects of ESG safeguards, sustainability rating systems, or methodologies that warrant deeper investigation.

Comments on the Quality of English Language

Minor editing of English language required

Author Response

The abstract suggests broad conclusions regarding the integration of sustainability indicators into ESG safeguards without specifying the extent of variability or inconsistency among different banks.

Comment addressed with new abstract in line 12.

Introduction

 1- The introduction is well written, but it needs further expansion, and it -also needs to clarify the study’s contribution to the growing literature, as it is not enough to mention the objectives of the study and consider it as a contribution.

Comment addressed on lines 62,78

2-Furthermore, authors should conclude the introduction with a small paragraph outlining the remaining sections of the study

Literature Review

 1- The review focuses on ESG safeguards and sustainability rating systems but could be enriched by integrating findings from broader literature on sustainable finance, development outcomes, or ESG impact assessment methodologies. Therefore, it is suggested that the authors incorporate more recent literature into the literature review to develop it further, such as:

  https://doi.org/10.1002/csr.2563

Comment addressed on lines 114,135,209

2-The review summarizes various sources, it could further critique the methodologies, biases, or limitations of the existing research. This would strengthen the case for the study's contribution to filling knowledge gaps.

Comment addressed on line 332

3- There is an absence of a theoretical framework

Methodology

 The rationale for choosing the specific methodologies (documentation analysis, high-level matrix, qualitative methods) could be more thoroughly justified in terms of how they best address the research questions.

Comment addressed from  line 367, and 486

Analysis

The discussion of the analysis of the results is limited to their narration only, without any possible contribution from the authors. The authors must develop and improve the discussion based on the empirical and theoretical literature.

Comment addressed on lines from 604 in advance, from 746, 805

Conclusion

  1-The conclusion could be strengthened by providing more concrete, actionable recommendations for MDBs, policymakers, or sustainability rating organizations on how to improve the integration of ESG safeguards.

Comment addressed on lines from 833,949

2-Suggestions for future research could be more specific, perhaps pointing to particular aspects of ESG safeguards, sustainability rating systems, or methodologies that warrant deeper investigation.

Round 2

Reviewer 1 Report

Comments and Suggestions for Authors

Thank you for the revised manuscript. My previous comments on the paper title, abstract, study aim, research question, acronyms, etc., have been addressed. Yet, some tables are still not linked to the main text, and further editorial checks are needed.

Comments on the Quality of English Language

Readability enhanced after the revision. Further checks are needed.

Author Response

Thank you very much for your comments. It is true that some tables were not properly referenced and needed to be improved. Below are the points that have been improved and can be found in the attached revised manuscript.

All the lists of tables have been standardised to "tables" instead of "tables no." As well has been done with "figures"

Table 6 has been cited on the line  445 "as shown in the table 6 below."

Table 8 has been cited on the line  545 "table 8 which illustrates the detailed scores from the High-Level Matrix by areas and credits, providing an in-depth view of how each bank or financial institution handles the various credits within their ESG safeguard policies."

Table 9 has been cited on the line  573 "As discussed in the previous paragraph in relation to the results achieved, table 9 below presents a percentage summary of the results obtained through the high-level matrix analysis, illustrating how the sustainability certification credits are distributed among the ESG thematic areas (environmental, social and governance) in each assessed institution."

Table 10 highlighted on the line 582

Table 11 cited on the line  721

Table 12 cited on the line 735

Table 13 cited on the line  742 "In the following table 13, we observe the same results presented in table 12 but based on ENVISION credits".

Table 14 cited on the line  780

Table 15 cited on the line  795

Table 16 cited on the line  805 "In the following table 16, we observe the same results presented in table 15 but based on CEEQUAL credits".

I hope that with these corrections you will be able to support the publication. Thank you very much for your review

Reviewer 2 Report

Comments and Suggestions for Authors

Dear Esteemed Author(s),

The revised manuscript enhanced its quality in a proper manner. The suggested changes and recommendations as formulated throughout the first peer-review round were considered and incorporated suitably. In its current form, the paper is worth publishing. Consequently, I endorse paper acceptance to publication.

Author Response

Thank you very much for your review and support of this article, your comments and recommendations have undoubtedly improved the article substantially.

Reviewer 3 Report

Comments and Suggestions for Authors

Thanks for the response and changes, I think the article has improved significantly.

Author Response

(The authors gave the same response as above.)

Reviewer 4 Report

Comments and Suggestions for Authors

As the assigned reviewer for this paper, I have thoroughly evaluated the revised version submitted by the authors. I am pleased to confirm that they have effectively addressed all the concerns raised during the initial review process.

Specifically, the authors have diligently implemented the suggested improvements, which have significantly enhanced the clarity, coherence, and overall quality of the manuscript. Moreover, they have successfully resolved the linguistic issues previously identified, ensuring that the text now meets the standards of academic excellence expected by the journal.

Therefore, I strongly recommend accepting this manuscript for publication in Sustainability.

Author Response

(The authors gave the same response as above.)
